# Germline-like TCR-α chains shared between autoreactive T cells in blood and pancreas

Peter S. Linsley [1] ✉, Maki Nakayama[2], Elisa Balmas[1], Janice Chen[1], Fariba Barahmand-pour-Whitman[1], Shubham Bansal [1], Ty Bottorff[1], Elisavet Serti [3], Cate Speake [1], Alberto Pugliese[4] & Karen Cerosaletti [1]

Human type 1 diabetes (T1D) is caused by autoimmune attack on the insulin-producing pancreatic beta cells by islet antigen-reactive T cells. How human islet antigen-reactive (IAR) CD4+ memory T cells from peripheral blood affect T1D progression in the pancreas is poorly understood. Here, we aim to determine if IAR T cells in blood could be detected in pancreas. We identify paired αβ (*TRA/TRB*) T cell receptors (TCRs) in IAR T cells from the blood of healthy, at-risk, new-onset, and established T1D donors, and measured sequence overlap with TCRs in pancreata from healthy, at risk and T1D organ donors. We report extensive *TRA* junction sharing between IAR T cells and pancreas-infiltrating T cells (PIT), with perfect-match or single-mismatch *TRA* junction amino acid sequences comprising ~29% total unique IAR *TRA* junctions (942/3,264). PIT-matched *TRA* junctions were largely public and enriched for *TRAV41* usage, showing significant nucleotide sequence convergence, increased use of germline-encoded versus non-templated residues in epitope engagement, and a potential for cross-reactivity. Our findings thus link T cells with distinctive germline-like TRA chains in the peripheral blood with T cells in the pancreas.

Many studies have investigated the role of islet antigen reactive (IAR) CD4+ and CD8 + T cells in peripheral blood of individuals with type 1 diabetes (T1D). IAR T cells have been investigated for their role in disease mechanisms and as therapeutic targets and biomarkers for β cell destruction[1–7]. Levels of IAR T cells may be increased in the pancreas especially during the active phases of islet autoimmunity, which may last months to years before and after clinical diagnosis[2,3]. Since pancreatic biopsy is not tenable in living humans, most efforts have focused on peripheral blood. IAR CD4+ and CD8 + T cells are found in blood of at-risk and T1D donors, but also often in healthy controls (HC) donor[8–10]. Although distinctive phenotypic properties of IAR T cells in T1D donors suggest their association with disease, because IAR T cells are rare in the blood[9], it remains difficult to ascribe a role for them in pathogenesis in the pancreas.

A defining feature of T cells is the expression of T cell receptors (TCRs) on their surface. T cells proliferate in response to TCR recognition of antigenic peptides, resulting in clonal expansion of a population of cells with identical TCR sequences at both the nucleic acid and protein sequence levels and the same antigen specificity[11]. We recently identified a population of expanded IAR T cells with restricted *TRA* junctions and germline-constrained antigen recognition properties[12]. In combination with our other studies[13], these previous studies suggest the possibility that expanded IAR T cell TCR sequences in the blood of T1D patients represent the ontogeny of T cell autoimmune responses during disease. While this suggestion is attractive for both fundamental and translational investigations in T1D, evidence that IAR T cells with these characteristics are found in the pancreas is lacking. Studies have characterized islet infiltrating

[1]Benaroya Research Institute at Virginia Mason, Seattle, WA, USA. [2]Barbara Davis Center for Childhood Diabetes, Department of Immunology and Microbiology, University of Colorado School of Medicine, Aurora, CO, USA. [3]Immune Tolerance Network, Bethesda, MD, USA. [4]Department of Diabetes Immunology & The Wanek Family Project for Type 1 Diabetes, Arthur Riggs Diabetes & Metabolism Research Institute, City of Hope, Duarte, CA, USA. ✉e-mail: plinsley@benaroyaresearch.org

## Table 1 | Summary of islet T cell donors

| Group[a] | Donor group | Number of donors | Gender (M:F) | Age (median, range) | Time from diagnosis (median, range) | HLA DRB1 | | | | | | Sample type | Sample source |
|---|---|---|---|---|---|---|---|---|---|---|---|---|---|
| | | | | | | DR4/DR4 | DR4/DR3 | DR4/other | DR3/DR3 | DR3/other | other/other | | |
| PIT[b] T cells | T1D | 14 | 5:9 | 19 y (3-28) | 2 y (0.3-15) | 0 | 4 | 5 | 1 | 2 | 2 | Pancreas | nPOD, ADI, VUMC/Pittsburgh |
| | 1 Aab+ | 4 | 2:2 | 21 y (19-29) | -- | 0 | 0 | 1 | 0 | 2 | 1 | Pancreas | nPOD |
| | non-T1D control | 9 | 6:3 | 30 y (18-34) | -- | 1 | 1 | 0 | 1 | 2 | 4 | Pancreas | nPOD, ADI, IIDP |
| IAR Cohort 1[c] | T1D | 12 | 8:4 | 27 y (10-40) | 3.6 y (1.6-7.1) | 3 | 2 | 7 | 0 | 0 | 0 | PBMC | BRI Registry, TrialNet |
| | newT1D | 24 | 13:11 | 19 y (12-34) | <100d | 4 | 6 | 9 | 0 | 5 | 0 | PBMC | ITN |
| | HC | 12 | 8:4 | 28 y (13-38) | -- | 0 | 2 | 10 | 0 | 0 | 0 | PBMC | BRI Registry |
| IAR Cohort 2 | newT1D | 11 | 5:5 (1 NA) | 19 y (12-28) | <100d | 1 | 5 | 3 | 0 | 2 | 0 | PBMC | ITN |
| | 1 Aab+[d] | 8 | 3:5 | 29 y (11-52) | -- | 0 | 0 | 4 | 2 | 2 | 0 | PBMC | TrialNet |
| | >1 Aab+[d] | 6 | 2:4 | 19 y (15-45) | -- | 2 | 1 | 1 | 0 | 1 | 1 | PBMC | TrialNet |
| | noAAb (HC) | 6 | 5:1 | 20 y (14-39) | -- | 0 | 1 | 2 | 0 | 2 | 1 | PBMC | BRI Registry |

a A more detailed description of individual donors is shown in Supplementary Data 1
b PIT pancreatic infiltrating T cells, T1D >1 year from T1D onset, newT1D <100 days from T1D onset, Aab autoantibody, nPOD Network for Pancreatic Organ Donors with Diabetes, ADI Alberta Diabetes Institute Islet Core, VUMC/Pittsburgh Vanderbilt University Medical Center/University of Pittsburgh, IIDP Integrated Islet Distribution Program, BRI Benaroya Research Institute, ITN Immune Tolerance Network.
c Donors in Cohort 1 were included in our previous studies (PMID: 37886513).
d 1 IAR AAb+ and >1 AAb+ donors were pooled together into a single AAb+ group for analysis

T cells in the pancreas from organ donors with T1D[14–16], but their link to IAR T cells in the blood is unclear.

In this study, we examine IAR T cells in the pancreas by cross-sectional comparisons of TCR junction sequences between circulating IAR T cells and pancreatic infiltrating T cells (PIT) from different donors. We identify and characterize a sizable fraction of IAR TCRs from peripheral blood with *TRA* junctions that share perfect- or single-mismatch protein sequences with PIT TCRs.

## Results

### Isolation of TCR sequences from IAR and PIT T cells

Our central hypothesis is that in vivo expansion of IAR T cells seen in peripheral blood[13] reflects autoimmune responses in the pancreas during T1D. This hypothesis predicts the presence of IAR T cells in the pancreas during disease. To test this prediction, we utilized the extreme sequence diversity of TCRs to enable their use as "barcodes" for clonal populations of T cells recognizing specific antigens[17]. Significant overlap between IAR and PIT cell TCR sequences would therefore be a prerequisite for a biological role for IAR TCRs in both blood and the pancreas.

We used TCRs from IAR T cells isolated from the peripheral blood of two cohorts (Table 1, Supplementary Data 1). *Cohort 1* was from our previous scRNA-seq comparisons of IAR T cells from HC donors ($n = 12$), new-onset T1D donors (newT1D) < 100 days from diagnosis ($n = 24$), and established T1D donors (T1D, $n = 12$)[12,13]. *Cohort 2* comprised additional HC (with no islet-directed autoantibodies) ($n = 6$) and newT1D donors ($n = 11$); donors with single autoantibodies (moderate risk for developing T1D, $n = 8$); and donors with multiple autoantibodies (high risk, $n = 6$). Patient characteristics are summarized in Table 1 and Supplementary Data 1.

We isolated IAR CD4 + T cells from these donors using a CD154 activation induced marker (AIM) assay, single-cell sorted them, and subjected them to paired TCR chain determination using scRNA-seq[12,13]. Paired in-frame αβ (*TRA/TRB*) IAR TCRs identified in single cells were subjected to several additional filtering steps, including removal of mucosa-associated invariant T cell (MAIT) and invariant natural killer T cell (iNKT) sequences before use in subsequent analyzes. TCRs analyzed are presented in Supplementary Data 2 and summarized in Table 2.

For PIT cells, we utilized paired TCRs identified by reverse transcription PCR of islets or pancreatic tissues from organ donors provided by the Network for Pancreatic Organ Donors with Diabetes (nPOD)[18]; a protocol at Vanderbilt University Medical Center/University of Pittsburgh; the Integrated Islet Distribution program (IIDP); and the Alberta Diabetes Institute Islet Core (ADI). Pancreatic donor characteristics are summarized in Table 1 and Supplementary Data 1. Paired in-frame αβ (*TRA/TRB*) PIT TCRs are listed in Supplementary Data 2 and summarized in Table 2. PIT cell TCR sequences contained multiple perfectly matched amino acid junction sequences from preproinsulin reactive TCRs found in islets and peripheral blood[19].

### TCRs from IAR T cells in blood share *TRA* chain sequence identity with PIT cell TCRs

For our initial experiments, we focused on *Cohort 1*. Approximately 90% of these donors had high-risk *DRB1*0401* HLA class II alleles, while ~10% had *DRB1*0301* alleles[12]. Using molecular cloning, lentiviral re-expression, and TCR functional assays, we previously identified specific islet antigen epitopes that triggered 29/47 (~62%) of TCRs from expanded clones tested from this cohort[12,13]. Reason(s) why the remaining 18 TCRs tested did not demonstrate peptide specificity remain unknown but may involve suboptimal avidity, and/or presentation by MHC class II molecules not tested. These TCRs tested for specificity, therefore, represented IAR T cells from peripheral blood with a wide range of specificities.

**Table 2 | Characteristics of the TCRs used in this study**

| Source | All junctions | | | Unique junctions | | |
|---|---|---|---|---|---|---|
| | TRA | TRB | Combined | TRA | TRB | Combined |
| **Cohort 1** | 2725 | 2590 | 5315 | 2174 | 2136 | 4310 |
| **Cohort 2** | 1182 | 1104 | 2286 | 1090 | 1051 | 2141 |
| **Total 1 + 2** | 3907 | 3694 | 7601 | 3264 | 3187 | 6451 |
| **PIT TCRs** | 6681 | 7317 | 13,998 | 4614 | 5143 | 9757 |

To test for TCR overlap, we tested for perfectly matched recombined V-J or V-D-J junction sequence overlap between IAR TCRs from *Cohort 1* with PIT cell TCRs. Although PIT TCRs represented a heterogeneous group of donors that varied among multiple parameters that may affect TCR repertoires, including cell type, patient group, and HLA class II allele (Supplementary Data 1, Supplementary Fig. 1), we elected to initially test all TCRs together. Unless otherwise noted, sequence comparisons were made at the amino acid level, as we reasoned that perfectly matched junctions were most likely to reflect conserved function. Multiple IAR junction sequences, mainly *TRA* junctions compared to *TRB* junctions ($n = 55$ and 7, respectively), perfectly matched pooled PIT junctions (Fig. 1A, Supplementary Fig. 1A). As expected, with PIT junctions, the same number of *TRA* than *TRB* junctions likewise perfectly matched IAR junctions (Fig. 1B). IAR junctions, which were isolated from CD4 + T cells, perfectly matched similar numbers of PIT junctions from both CD4+ and CD8 + T cells (Fig. 1C, Supplementary Fig. 1B). TCR chain sharing between CD4+ and CD8 + T cells has been noted previously, more frequently between individual *TRA* than *TRB* chains[20]. Overall, *Cohort 1* had ~2.5% (55/2,174) of total unique IAR T cell *TRA* junctions that perfectly matched PIT junctions. Conversely, ~0.56% (55/9,757) of PIT cell TCRs matched IAR T cell TCRs. The distribution of IAR perfect matches between PIT *TRA* and *TRB* junctions (Fig. 1A) differed significantly from the distribution of *TRA* and *TRB* junctions in total PIT TCRs (Supplementary Fig. 1) (p-value = 1.1e-11, Fisher's exact test). In contrast, distributions of IAR perfect matches by donor group (HC, T1D, and newT1D), cell type (CD4 + , CD8 + ), or HLA class II alleles did not differ significantly from the distributions in total PIT TCRs (Supplementary Fig. 1C and D) (all p-values > 0.05, Fisher's exact test). These findings supported our decision to maximize power by utilizing combined PIT TCRs for most analyzes.

### PIT-matched junctions are enriched in IAR CD4 + T cells relative to unselected TCR repertoires

We hypothesized that junction sequence matching between IAR and PIT TCRs was greater than would occur by chance in unselected repertoires. An unselected repertoire determined by the same technology from healthy individuals matched by donor characteristics and HLA type to IAR T cell donors was not available. We selected instead two PBMC repertoires from Su et al. (16 uninfected (HC) and 129 donors with COVID-19, comprising 2513 and 198,753 unique junctions for HC and COVID-19 patients, respectively)[21]. It should be noted that these TCRs were determined by different technologies (a 5′RACE method[21] versus the RT-PCR method optimized for rare cells that was used in our data set[12,13]), and were not matched by donor characteristics and HLA type to IAR T cell donors.

We used Fisher's exact test to compare numbers of perfectly PIT-matched to non-matched *TRA* junctions from IAR T cells versus unselected HC and COVID-19 T cells. We found that a higher fraction of PIT junction matches with IAR *TRA* junctions than with junctions from unselected repertoires: there were 55 perfect matches with IAR *TRA* junctions (2.5%) versus 154 (1.4%) with HC junctions ($n = 11,274$) and 1342 (0.7%) for COVID patient junctions ($n = 198,753$). These comparisons yielded $\log_2$ odds ratios >1 (or >2-fold in linear units), outside the 95% confidence intervals (Fig. 1D) (p-values 1.9e-4 and 8.5e-16, respectively, by Fisher's exact test). In contrast, the $\log_2$ odds ratio was

<0 (or 1 in linear units), for PIT-matched to non-matched junctions from COVID-19 patients versus HCs donor(Fig. 1D) (p-value > 0.05). Thus, there was a higher fraction of PIT TCR perfect matches with IAR *TRA* junctions than unselected human TCR repertoires from HC and COVID-19 patients. In contrast, PIT TCRs did not show greater numbers of perfect matches with IAR *TRB* junctions (Fig. 1E).

TCRs from IAR T cells with mismatched junctions may also share functional properties. While we expect that shared function would be more likely with mismatches that conserve amino acid electrochemical properties, we have ignored these properties to simplify subsequent analyzes. We tested for enrichment of IAR T cell *TRA* chains in PIT TCRs over a range of mismatch values by calculating pairwise Levenshtein index values. A plot of the numbers of PIT *TRA* junction matches with IAR versus HC T cells at different PIT mismatch thresholds (Fig. 1F) showed significant off-diagonal skewing in the direction of IAR *TRA* junctions (slope = 0.872, p-value (that the slope was not equal to 1) = 3e-3, by linear modeling). Thus, enrichment of PIT-matched to non-matched *TRA* junctions versus unselected HC TCRs was not limited to perfect sequence matches. Ninety-five % confidence intervals for PIT junction sequence overlaps with 0 and 1 mismatches showed greatest divergence between IAR T cell and HC TCRs. Taken together, we identified 582 unique *TRA* chains from a total of 2174 unique *TRA* junctions (~27%) IAR CD4 + T cells in *Cohort 1* with perfect or single mismatches with PIT TCR junctions. We did not observe significant off-diagonal divergence of *TRB* junctions that were paired with PIT-matched *TRA* junctions, indicating more highly mismatched sequences (Fig. 1G). Although there were no perfect PIT-matches to IAR TCRs of known specificity, selected IAR TCRs of unknown specificity with perfect PIT matches are shown in Table 3. There were, however, also numerous IAR TCRs with known specificity having single mismatches to PIT TCRs (Table 4). With both perfect and single PIT mismatches, the paired *TRB* junctions were markedly more divergent than the matched *TRA* junctions.

### PIT junction matching in a second cohort of IAR T cells

Since the results in Fig. 1A–I were obtained from only a single donor cohort (*Cohort 1*), we did not know how broadly applicable our findings were. To examine this question, we repeated the analyzes with an independent cohort of IAR T cells (*Cohort 2*) (Table 1 and Supplementary Data 1). These analyzes yielded similar results (Supplementary Fig. 2). There were again more *TRA* junctions compared to *TRB* junctions ($n = 30$ and 4 *TRA* and *TRB*, respectively) showing perfect matches with total PIT junctions (Fig. S2A, B), spread between CD4+ and CD8+ cells (Fig. S2C). There also were more *TRA* than *TRB* junctions showing 0 or 1 mismatches with PIT junctions (Supplementary Fig. 2D–G). In total, there were 360 unique *TRA* chains from *Cohort 2* with perfect matching or single mismatches with PIT TCR junctions in a total of 1,090 unique *TRA* junctions (~33%). These findings demonstrated that our observations with *Cohort 1* were not restricted to a single data set and therefore had a potentially broader range of islet specificities. To increase power in subsequent analyzes, we combined TCRs from *Cohorts 1* and *2*, yielding 3264 and 3187 unique *TRA* and *TRB* junctions (Table 2). In both *Cohorts 1* and *2*, there were more PIT-matches with IAR *TRA* junctions than *TRB* junctions (Fig. 1, Supplementary Fig. 1, and Supplementary Fig. 2). Since the combined data were more optimally powered to analyze *TRA* than *TRB* junctions, we chose to focus on them in later experiments. In combination, there were 942 unique *TRA* junctions with perfect- or single mismatches to PIT junctions, from a total of 3264 unique IAR *TRA* junctions (~29%). We hereafter refer to these junctions as "PIT-matched".

### PIT *TRA* junction matches extend to include the *J* gene but not the *V* gene

To test the extent of sequence identity of PIT-matched IAR *TRA* chains, we extended the requirement for sequence matches into the

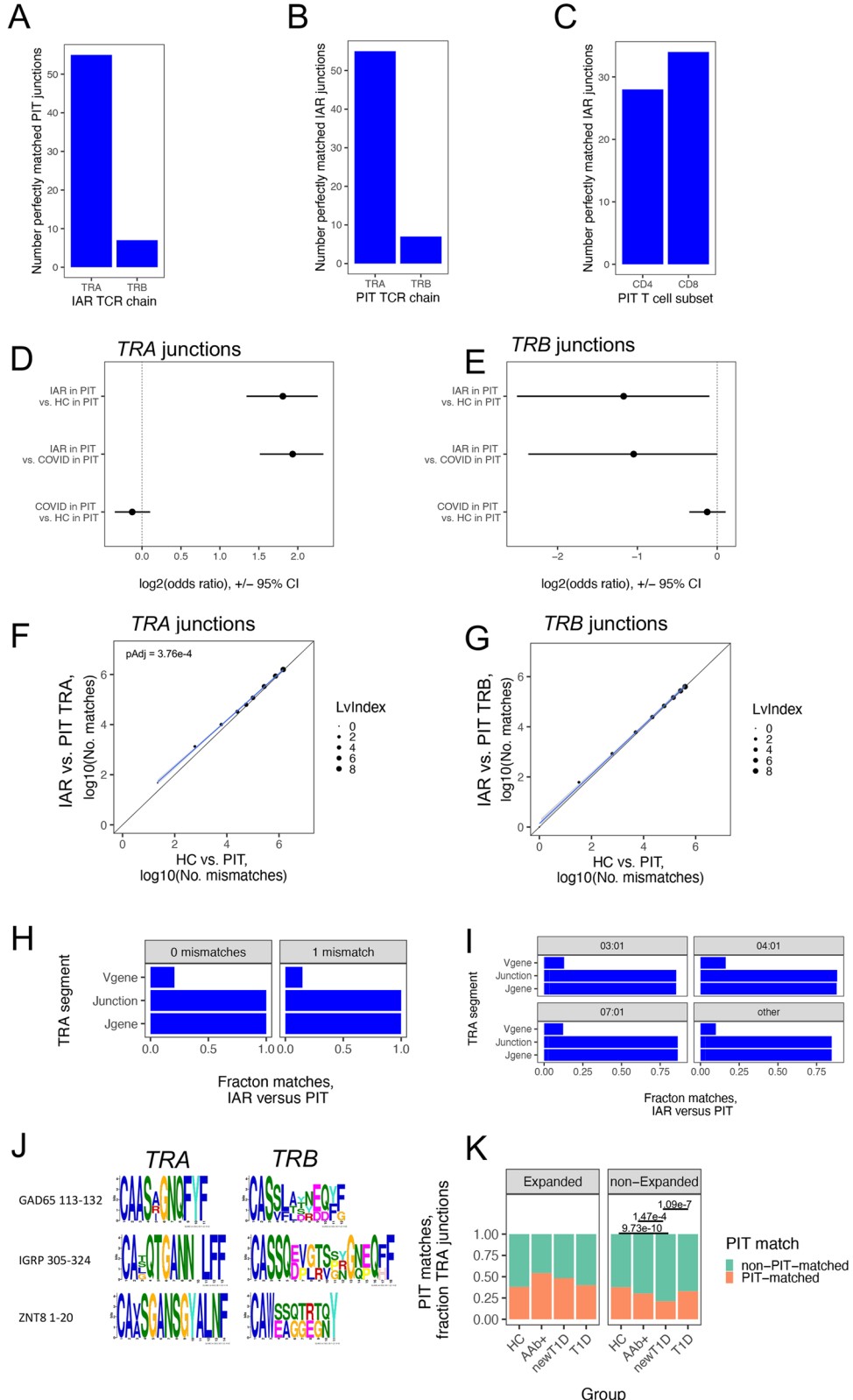

V and J gene segments flanking matched junctions (Fig. 1H). Since the V and J gene segments in TCRs are much longer than junctions[22], they are less amenable to string match comparisons, prompting us to consider V and J genes as identical if they simply had the same name. For all junctions with 0 or 1 PIT mismatches, sequence identity extended towards the C terminus to include use of identical J genes (Fig. 1H). In contrast, sequence identity towards the N terminus was less marked, with only ~20-25% of matched junctions having identical V genes.

Since the TCR V gene contains the CDR1 and CDR2 regions that contact the MHC class I or class II molecules[23], it was possible that the low frequency of V gene identical matches reflected peptide presentation to PIT TCRs by different HLA molecules. IAR T cell TCRs were associated primarily with high-risk HLA genotypes[12], whereas PIT TCRs

**Fig. 1 | IAR T-PIT TCR sequence matching.** Panels A–G utilized TCRs from *Cohort 1*; panels H-K, TCRs from combined Cohorts 1 + 2. A) Perfectly matched IAR *TRA* and *TRB* junctions from *Cohort 1* (Table 2) in PIT TCRs (n = 9,757 unique *TRA* and *TRB* junctions, Table 2). **B** Perfectly matched PIT *TRA* and *TRB* junctions in IAR TCRs (n = 4,310 unique *TRA* and *TRB* junctions). **C** Perfectly matched PIT T cell TCR junctions from CD4+ and CD8+ cells in IAR TCRs. **D** Perfectly PIT-matched versus non-matched *TRA* junctions from *Cohort 1* IAR T cell TCRs (n = 55 matched and 2119 non-matched) versus *TRA* junctions from HC[20] (185 matched and 24,968 non-matched) and COVID-19 patients[20] (1341 matched and 197,412 non-matched). **E** As in D but using *TRB* junctions (n = 7 matched and 2129 unmatched). **F** PIT junction matching in IAR *TRA* junctions versus junctions from HC donors[20]. Dot sizes, Levenshtein index values. Diagonal line, equivalency line. Blue line, best fit line from linear modeling. The pAdj for a slope different from 1 was calculated using linear

modeling and ANOVA. Gray shading, 95% confidence intervals. **G** As in F but using *TRB* junctions. **H** PIT matching by different *TRA* chain segments in combined *Cohorts 1* and *2* (n = 3264 unique *TRA* junctions). Lv0 and Lv1, Levenshtein index values of 0 and 1 (n = 74 and 2025 unique *TRA* junctions, respectively). **I** PIT matching by *TRA* junction segments in combined *Cohorts 1* and *2* according to HLA-DRB1 alleles in PIT donors. There were: n = 484, 153, 199, and 95 junctions from donors having 03:01, 04:01, 07:01, and other HLA-DRB1 alleles, respectively. **J** Junction sharing by TCRs with PIT-matched *TRA* junctions. One TCR in each group recognized[13] the indicated autoantigen epitope. **K** Total *TRA* chains (Table 2) in expanded (n = 1000) and non-expanded (n = 2907) cells from combined *Cohorts 1* and *2* were tested for differences in PIT-matched and non-PIT-matched junctions from HC, Aab + , newT1D, and T1D donors using two-sided Fisher's exact tests.

## Table 3 | Examples of PIT TCRs sharing perfectly matched *TRA* chains with IAR T cells. [A]

| TCRs *TRA* chain | | | | *TRB* chain | | |
|---|---|---|---|---|---|---|
| Cells | V gene | Junction | J gene | V gene | Junction | J gene |
| IAR | TRAV1-2 | CAVRMNTGFQKLVF | TRAJ8 | TRBV11-2 | CASSFGGGATDTQYF | TRBJ2-3 |
| PIT | TRAV1-2 | CAVRMNTGFQKLVF | TRAJ8 | **TRBV9** [B] | CASS**VG**M**DPGLGYNEQF**FF | **TRBJ2-1** |
| IAR | TRAV12-1 | CVVNDQAGTALIF | TRAJ15 | TRBV7-2 | CASSLDAGRNSPLHF | TRBJ1-6 |
| PIT | TRAV12-1 | CVVNDQAGTALIF | TRAJ15 | **TRBV20-1** | C**SARGYNSYEQY**F | **TRBJ2-7** |
| IAR | TRAV12-1 | CVVQGGSYIPTF | TRAJ6 | TRBV5-4 | CASSLVTSGENEQFF | TRBJ2-1 |
| PIT | TRAV12-1 | CVVQGGSYIPTF | TRAJ6 | ND [C] | | |
| IAR | TRAV12-2 | CAVNQAGTALIF | TRAJ15 | TRBV28 | CASSFGSGADYGYTF | TRBJ1-2 |
| PIT | TRAV12-2 | CAVNQAGTALIF | TRAJ15 | **TRBV29-1** | C**SVFDWDRGPGELF**F | **TRBJ2-2** |
| IAR | TRAV12-2 | CAVRSNFGNEKLTF | TRAJ48 | TRBV19 | CASGTDSY-EQYF | TRBJ2-7 |
| PIT | TRAV12-2 | CAVRSNFGNEKLTF | TRAJ48 | **TRBV28** | CAS**RTTGGTE**AFF | **TRBJ1-1** |
| IAR | TRAV13-1 | CAASIGTGTASKLTF | TRAJ44 | TRBV9 | CASSVA-GGGY-EQYF | TRBJ2-7 |
| PIT | TRAV13-1 | CAASIGTGTASKLTF | TRAJ44 | **TRBV24-1** | CAT**SDPS**GGGG**NEQF**F | **TRBJ2-1** |
| IAR | TRAV41 | CAASNTGNQFYF | TRAJ49 | TRBV28 | CAIGGRVYNEQFF | TRBJ2-1 |
| PIT | TRAV41 | CAASNTGNQFYF | TRAJ49 | **TRBV5-1** | CAS**SGSNYGYT**-F | **TRBJ1-2** |
| IAR | TRAV5 | CAERGLTGGGNKLTF | TRAJ10 | TRBV9 | CASSVGGDFYNEQFF | TRBJ2-1 |
| PIT | TRAV5 | CAERGLTGGGNKLTF | TRAJ10 | **TRBV12-5** | CAS**GLTRGSTDT**QYF | **TRBJ2-3** |
| IAR | TRAV8-2 | CVVSGGSNYKLTF | TRAJ53 | TRBV29-1 | CSAHGGGGT--EAFF | TRBJ1-1 |
| PIT | TRAV8-2 | CVVSGGSNYKLTF | TRAJ53 | **TRBV6-1** | CAS**SQG**T**PQYNEQF**FF | **TRBJ2-1** |
| IAR | TRAV8-3 | CAVGPTGTASKLTF | TRAJ44 | TRBV7-6 | CASSTNHQ------ETQYF | TRBJ2-5 |
| PIT | TRAV8-3 | CAVGPTGTASKLTF | TRAJ44 | **TRBV3-1** | CASS**GTGTGGLSPQ**ETQYF | **TRBJ2-5** |

[A]Shown are amino acid sequence comparisons of a selected subset (n = 10 of the total of 55) of perfectly matching IAR and PIT T cell TCRs of unknown specificity. Dashes indicate gaps.
[B]Bold font indicates a mismatch with the IAR reference sequence (top row of each pair).
[C]*ND* not determined

were from individuals having a wider variety of HLA genotypes. We, therefore, broke down the PIT-matching data from Fig. 1H by HLA class II *DRB1* genotype. Individuals having 03:01, 04:01, 07:01 and pooled other *DRB1* genotypes all showed similar frequencies of *V gene* identical matches (Fig. 1I). Thus, IAR TCRs (from ~90% HLA DRB1-04 donors)[12] showed neither preferential matches with PIT junctions from donors with different class II alleles (Supplementary Fig. 1D), nor did they show class II allele-dependent differences in the extent of matches in to *V gene* segments (Fig. 1I). Together, these findings suggest that peptide presentation by different PIT HLA molecules did not have a major effect on matching with IAR *TRA* junctions. Other studies have noted HLA-independent associations with *TRB* chains[24].

### PIT-matched *TRA* junctions are paired with diverse *TRB* chains

Although we observed a few *TRB* perfect matches (Fig. 1A), these were not paired with perfectly matched *TRA* chains. Likewise, we observed that perfect or single mismatch PIT-matched *TRA* chains were paired with quite different *TRB* chain sequences. This was true with TCRs of both unknown (Table 3) and known[12] (Table 4, and Fig. 1J) specificity. The divergence of *TRB* sequences with perfect or single mismatch PIT-

matched *TRA* junctions makes it uncertain that these TCRs share epitope specificity. We conclude that PIT-matched *TRA* chains were paired with diverse *TRB* chains.

### PIT *TRA* matches with IAR T cell junctions were expanded early in disease

We speculated that PIT *TRA* matches were related to T cell expansion and T1D progression. As a test, we quantified levels of PIT *TRA* matches in IAR TCR junctions along the continuum of stages during the T1D disease process[25], using combined *Cohorts 1* and *2* (Fig. 1K). Since we were able to sample relatively few 1AAb and 2AAb individuals in Cohort 2, we combined these donors into a single category (termed AAb + ). We then compared levels of PIT-matched *TRA* junctions from expanded (>1 cell) and non-expanded (1 cell only) T cell clones. In cells with expanded TCRs, we saw an increase in the fraction of PIT-matched junctions from Donors in both AAb+ and newT1D donors, then a decrease in T1D donors to a similar level as in HCs donor(Fig. 1K). These comparisons were underpowered with respect to both donor and cell numbers and group differences did not reach significance after multiple correction. In contrast, there were strongly significant

**Table 4 | PIT TCRs sharing a single mismatched *TRA* chain with IAR T cells of known specificity. [A]**

| TRA chain | | | | TRB chain | | |
|---|---|---|---|---|---|---|
| Specificity | *V gene* | Junction | *J gene* | *V gene* | Junction | J gene |
| GADp15 | TRAV41 | CAAA-GNQFYF | TRAJ49 | TRBV12-4 | CASSFT--YNEQFF | TRBJ2-1 |
| ND[C] | **TRAV29/DV5**[B] | CAA**R**-GNQFYF | TRAJ49 | **TRBV6-2** | CASS**LLNLD**NEQFF | TRBJ2-1 |
| ND | **TRAV21** | CAA**I**-GNQFYF | TRAJ49 | **TRBV29-1** | C**SVLRDRASY**EQYF | **TRBJ2-7** |
| ND | **TRAV29/DV5** | CAA**SA**GNQFYF | TRAJ49 | **TRBV4-1** | CASS**LAATRDDYGYT**F | **TRBJ1-2** |
| IGRP39 | TRAV25 | CAGQTGANNLFF | TRAJ36 | TRBV4-3 | CASSQEVGTVPNPQPHF- | TRBJ1-5 |
| ND | **TRAV16** | CAL**Q**TGANNLFF | TRAJ36 | | | |
| ND | **TRAV13-1** | CA**T**QTGANNLFF | TRAJ36 | TRBV9 | CASS**VGR---SS**YNEQFF | TRBJ2-1 |
| ND | **TRAV24** | CA**S**QTGANNLFF | TRAJ36 | **TRBV4-1** | CASS**QDPLTSGRG**NEQF**F** | TRBJ2-1 |
| ZNP1 | TRAV13-1 | CAASGANSGYALNF | TRAJ41 | TRBV30 | CAWSAQGETQYF | TRBJ2-5 |
| ND | **TRAV8-4** | CA**V**SGANSGYALNF | TRAJ41 | TRBV30 | CAW**ESGTRGNYGYT**F | **TRBJ1-2** |
| Multiple[D] | TRAV29DV5 | CAASRYSGGGADGLTF | TRAJ45 | TRBV12-4 | CASSPQGGNTEAFF | TRBJ1-1 |
| Multiple | TRAV29DV5 | CAASRYSGGGADGLTF | TRAJ45 | TRBV12-4 | CASS**V**QGGNTEAFF | TRBJ1-1 |
| ND | **TRAV13-1** | CAASR**S**SGGGADGLTF | TRAJ45 | ND | ND | ND |
| Multiple | TRAV23DV6 | CAASNPDYKLSF | TRAJ20 | TRBV5-1 | CASSFTEGNTEAFF | TRBJ1-1 |
| ND | **TRAV13-1** | CAASN**N**DYKLSF | TRAJ20 | TRBV28 | CASS**GRAD--EQ**FF | **TRBJ2-1** |

[A]Shown are amino acid sequence comparisons of PIT TCRs having single mismatched *TRA* junctions with IAR TCRs of known specificity. The specificity of IAR TCRs is specified.
[B]Bold font indicates a mismatch with the IAR reference sequence.
[C]*ND* not determined.
[D]Multiple, IAR TCR with multiple specificities.

decreases in PIT-matched *TRA* junctions in the more numerous non-expanded TCRs that were the opposite of the expanded *TRA* junctions, i.e. higher numbers of non-expanded PIT-matched *TRA* junctions in HC and T1D compared to AAb+ and newT1D (Fig. 1K).

To control for p-value inflation by increased numbers of junctions from relatively few donors, we repeated this analysis at the donor level and observed the same trends as with junction frequency (Supplementary Fig. 3). The effects at the donor level were weaker, suggesting that these studies were underpowered. Together, these analyzes suggest that an elevation of expanded PIT-matched *TRA* chains in blood occurred prior to the onset of clinical disease.

**Publicity, generational probability, and TCR convergence of PIT-matched TCRs**

Since different cohorts were used to isolate IAR and PIT TCRs, PIT-matched *TRA* junctions were shared between different donors (public). We hypothesized that PIT-matched TCRs were related to the public TCRs with diverse specificities and germline-constrained antigen recognition properties that we previously described in IAR T cells[12]. To test this hypothesis, we asked whether PIT-matched *TRA* and associated *TRB* junctions were enriched with public and private junctions. Public IAR and PIT-matched *TRA* junctions showed significant overlap, whereas private IAR *TRA* junctions and PIT-matched *TRA* junctions did not (Supplementary Fig. 4A, B). Supplementary Fig. 4 Public IAR *TRB* junctions showed weak overlap with PIT-matched *TRB* junctions, but not private IAR *TRB* junctions did not (Supplementary Fig. 4C, D). Thus, there was a strong overlap between *TRA* junctions from public TCRs and PIT-matching *TRA* junctions.

Public TCR sharing may result from the expansion of clones with a high probability of recombination[26]. The probability of generating individual recombination events may be estimated by the generation probability (*Pgen*) of junction sequences during TCR recombination[26]. To estimate the generational probabilities of PIT-matched TCRs, we calculated their *Pgen* values (Supplementary Fig. 5). *Pgen* values were higher overall (i.e., a higher propensity for generation by chance) for *TRA* than *TRB* junctions, presumably reflecting the greater numbers of random non-templated nucleotides in *TRB* junctions[27]. -log10(*Pgen*) values for both PIT-matched *TRA* (Fig. S5A), and their associated *TRB* junctions (Fig. S5B), were significantly higher than for non-PIT-

matched junctions, especially for *TRA* junctions (median -log10(*Pgen*) values of 7.9 and 9.2, respectively). The differences between PIT-matched and non-PIT-matched TCRs were smaller than the overall spread of *Pgen* scores (>10 orders of magnitude), suggesting heterogeneity in generational probabilities of different PIT-matching sequences.

Public TCR AA sequences TCRs may arise from use of identically rearranged nucleotide (nt) sequences across donors or from post-recombination antigen selection of clones with identical protein sequences encoded by different nt sequences (TCR convergence)[28–30]. To distinguish these possibilities, we examined TCR convergence in PIT-matched junctions (Supplementary Fig. 6). A total of 93/6451 IAR junctions (69 *TRA* and 24 *TRB*, 1.1%) were convergent (Supplementary Fig. 6A). This number is likely an underestimate because of the relatively few cells sampled and the resulting shallowness of the repertoire estimation. Significantly more *TRA* than *TRB* junctions were convergent (p-value = 7.4e-7, Fisher's exact test) (Supplementary Fig. 6A). Nearly all public (Supplementary Fig. 6B) but only a few private *TRA* junctions (Supplementary Fig. 6C) were convergent, despite private *TRA* junctions being found in more cells (more expanded) than public junctions (Supplementary Fig. 6D). This argues against sequencing errors as an alternative explanation for different nt sequences. Finally, convergent *TRA* junctions were more enriched with PIT-matched sequences than convergent *TRB* junctions (Supplementary Fig. 6E and Supplementary Fig. 6F). Thus IAR TCRs, particularly public *TRA* junctions, showed evidence of TCR convergence. Taken together, results from this section demonstrate that PIT-matched TCRs were enriched with public sequences, have high generational probability, and show TCR convergence.

**PIT-matched *TRA* junctions were more germline-like, and more hydrophobic than non-PIT-matched junctions**

To test for molecular differences between PIT-matched and non-matched junctions, we compared sequence features that have been correlated with TCR autoreactivity[12,31,32](Fig. 2). We utilized density plots for these comparisons to emphasize the range of TCR junction lengths. We found that the distribution of *TRA* junctions was significantly left shifted (shorter) in PIT-matched junctions than in non-matched junctions (p-value < 1e-4, Kolmogorov–Smirnov test)

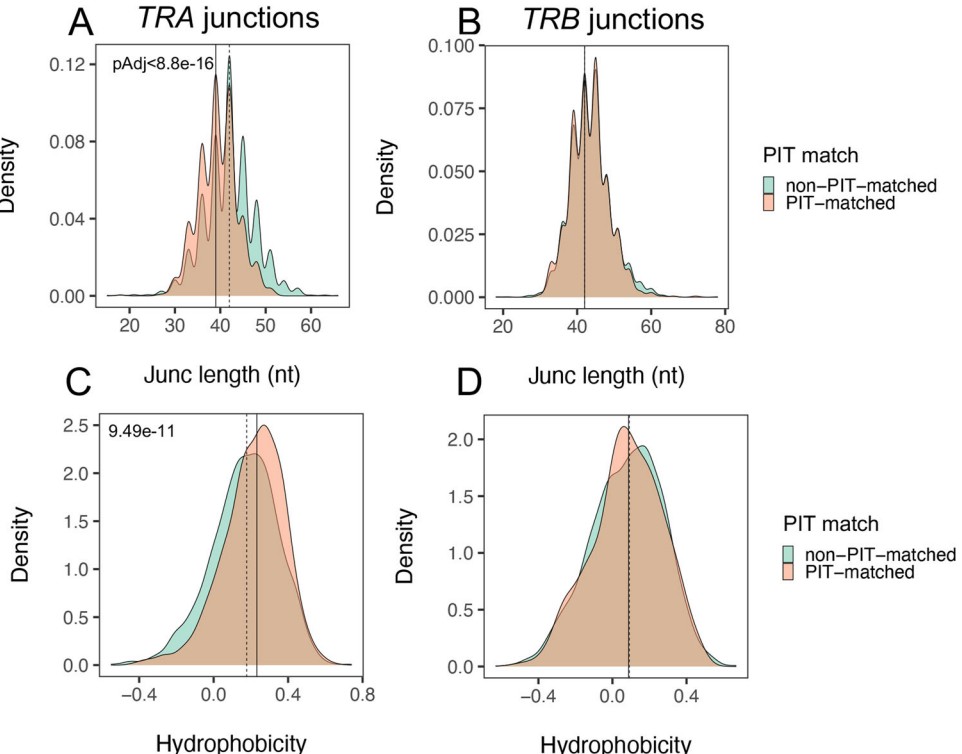

**Fig. 2 | PIT-matched *TRA* junctions are shorter and more hydrophobic than non- PIT-matched junctions.** Distributions of sequence lengths (in nt) for features of unique PIT-matched (*n* = 942 and 832) and non-matched (*n* = 2322 and 2355) *TRA* and *TRB* junctions, respectively, as delineated by IMGT/HighV-QUEST[61] from combined *Cohorts 1 + 2*. **A**, **B** junction lengths (Junc); **C**, **D** hydrophobicity. The significance of differences between PIT-matched (teal) and PIT-non-matched (peach) junctions was assessed using two-sided Kolmogorov-Smirnov tests. Solid vertical line, median value from PIT-matched junctions; dashed vertical line, median value from PIT-non-matched junctions.

(Fig. 2A). PIT-matched *TRA* junctions had a median length of ~39 nt, versus a median length of ~42 nt for PIT-non-matched junctions (i.e., 3 nt or 1 AA residue difference) (Fig. 2A). We did not see a significant difference in length with paired *TRB* junctions; *TRB* junctions paired with both PIT-matched and non-matched *TRA* junctions had identical median length values of ~42 nt (Fig. 2B). Furthermore, *TRA* junctions (Fig. 2C), but not paired *TRB* junctions (Fig. 2D), were more hydrophobic in PIT-matched versus non-matched TRA junctions (p-value < 1e-4). PIT-matched and non- matched *TRA* junctions had median hydrophobicity values of 0.23 and 0.18, respectively, on the Eisenberg hydrophobicity scale[33], falling between values for proline and tyrosine (0.12 and 0.26, respectively). Thus, PIT-matched *TRA* junctions, but not their paired *TRB* junctions, were shorter and more hydrophobic than PIT-non-matched *TRA* chains.

**Length differences between PIT-matched and non-matched *TRA* junctions map to peptide-contact regions**

We hypothesized that sequence features of PIT-matching *TRA* junctions were keys to the function(s) of IAR T cells having these TCRs. We first determined the positions of single AA mismatches between PIT and IAR T cell *TRA* junctions (*n* = 927). We found that mismatched residues were distributed with a mode at amino acid position 3 (Supplementary Fig. 7A). By comparison, the C terminal AA from *V genes* (3' end) and the N terminal AA of *J genes* (5' end), which mark the boundaries of germline-encoded residues in the respective gene segments, were centered at AA residues 3 and 5, respectively (Supplementary Fig. 7A). Considering individual IAR *TRA* junctions with single PIT mismatches, we observed that ~74% (687/927) AA mismatches were located at the C-terminus of the *V gene* or between the C-terminus of the *V gene* and the N-terminus of the *J gene* (Supplementary Fig. 7B). Thus, PIT mismatches occurred

in a mixture of germline and non-germline-encoded residues in the V-J recombination region.

We next focused our comparisons on regions important for TCR binding and function. TCR complementary determining regions (CDRs) convey the specificity for antigen and major histocompatibility complex molecules[34]. A comparison of CDR1 region lengths showed that the median CDR1 length was identical in PIT-matched and non-matched junctions, but the distribution was left-shifted significantly for PIT-matched junctions (Supplementary Fig. 8A). CDR2 regions showed no little difference between PIT-matched and non-matched *TRA* chains (Supplementary Fig. 8B), whereas CDR3 regions, as expected from the junction sequences comparisons in Fig. 2, were shorter in in PIT-matched *TRA* chains (Supplementary Fig. 8C), by ~3 nt (1 AA).

We wished to determine the source(s) of junction length differences between PIT-matched and non-matched *TRA* junctions. We compared N region lengths between the two groups of *TRA* chains and found that PIT-matched junctions had median N region lengths of ~3 nt (1 AA) whereas PIT-non-matched junctions had median N region lengths of ~5 nt (1-2 AA) (Supplementary Fig. 8D). We also quantified contributions of the *V gene* and *J gene* segments adjacent to the N region. The 3' end of the V region was slightly shorter in PIT-matched than non-matched *TRA* junctions (10 versus 11 nt) (Supplementary Fig. 8E), whereas the 5' end of the J region was slightly longer (27 versus 25 nt) (Supplementary Fig. 8F). This is consistent with our observed sharing of *J genes* but not *V genes*. Thus, the difference in CDR3 lengths was a complex product of genome- and non-genome-encoded regions in PIT-matched junctions. Framework (FR) regions FR1, FR2, and FR3 showed more modest differences in length between PIT-matched and non-matched *TRA* chains (Supplementary Fig. 8G–I), demonstrating some selectivity of the differences in CDR1 and CDR3 lengths.

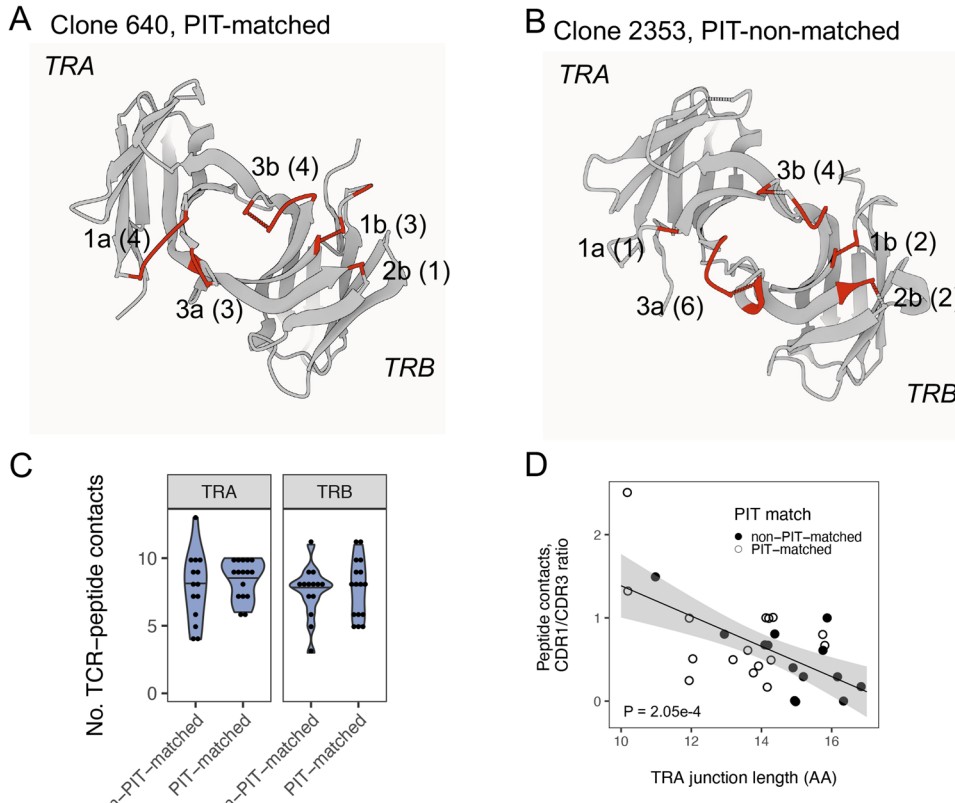

**Fig. 3 | Increased ratios of germline-encoded to recombined peptide contacts in PIT-matched TCRs. A** Gaussian Surface representation of a model of Clone 640, a PIT-matching TCR with a 10 AA *TRA* junction (7 CDR contacts) paired with a 12 AA *TRB* junction (8 CDR contacts), that binds the GAD65 113-132 peptide presented by the HLA DRB1*0401 class II molecule[12]. The viewing plane is from the interface with the MHC molecules, which together with the peptide, have been removed from the representation for clarity. Labels indicate the CDR loops, with letters denoting the chain (α or β); and numbers, the CDR loop. Numbers in parentheses are the number of predicted peptide contacts with the CDR loop. **B** A model of Clone 2353, a PIT-non-matching TCR with a 17AA *TRA* junction (7 CDR contacts) paired with a 14 AA *TRB* junction (8 CDR contacts), that binds the IGRP305-324 peptide presented by the HLA DRB1*0401 class II molecule. **C** The number of total peptide contacts for either the *TRA* or *TRB* chains did not differ significantly between PIT-matched and PIT-non-matched TCRs (p-value > 0.05, two-sided Wilcoxon signed rank test). Each dot represents a value for an individual donor. The width of the violins represent frequency; horizontal lines within the violins represent median values. **D** Ratios of *TRA* CDR1 to CDR3 peptide contacts decrease with increasing *TRA* junction length. The *P* value is for a null hypothesis of a slope of 0, calculated using linear modeling. Gray shading, 95% confidence intervals.

Together, these analyzes showed that PIT-matched and non-matched *TRA* junctions differed regions important for peptide binding, including templated *V gene* region CDR1 and both templated and non-templated regions of CDR3..

We also wished to determine whether there were *TRA V gene* differences between PIT-matched and non-matched *TRA* junctions. We thus compared enrichment of *V gene* segments in PIT-matched versus non-matched TCRs. This comparison identified *TRAV41*01 V genes* as significantly over-represented (pAdj -1e-4) in PIT-matched IAR T cell *TRA* chains (Supplementary Fig. 9A). In contrast, we did not identify any *V genes* as significantly over- or under-represented (pAdj >0.05) in PIT-matched IAR T cell *TRB* chains (Supplementary Fig. 9B). *TRAV41*01 V genes* were previously identified, together with MAIT and iNKT TCRs, in innate T cells having invariant *TRA* (iTRA) chains[35]. The over-represented *V gene*, *TRAV41*01*, had a CDR1 region length of 15 nt (5 AA), compared to the median of 18 nt (6 AA), thus contributing to the overall shorter CDR1 regions in PIT-matched *TRA* junctions.

### Determining predicted peptide binding contacts for PIT-matching and non-matching *TRA* chains

CDR1 and CDR3 regions are involved in binding to peptides presented by MHC molecules[23], leading us to hypothesize that the observed length variation of these regions suggested altered peptide binding properties for PIT-matched IAR *TRA* chains. As a test, we used TCRmodel2[36] to predict peptide binding residues in tri-molecular structures of PIT-matched and PIT-non-matched TCRs complexed with peptide-class II MHC molecules. For model input, it was desirable to use TCRs with cognate peptides. We therefore selected paired *TRA* and *TRB* sequences from a set of 30 IAR CD4+ TCRs, largely from expanded clones, with known specificity (16 PIT-matched and 14 PIT-non-matched TCRs), together with their cognate peptides[12]; and sequences of HLA *DRA*01:01* and *DRB1*0401* MHC class II subunits[37].

From best fit models for each TCR, we identified likely peptide contact residues (TCR residues <5 Å in distance from the bound peptide chain) and mapped these to *TRA* and *TRB* sequence features. We first visually compared TCR-peptide contact residues in representations of PIT-matched and PIT-non-matched TCRs with extremes in *TRA* junction length (Fig. 3A, B). This comparison showed that IAR Clone 640 (10 AA, PIT-matched), specific for GAD65 113-132, had more predicted peptide contacts that mapped to CDR1, and fewer that mapped to CDR3, than IAR Clone 2353 (17 AA, PIT-non-matched), specific for IGRP 305–324. To extend these observations to the larger data set, we compared the numbers of TCR contacts of each chain from the set of 30 IAR TCRs (Supplementary Data 3). We displayed the numbers of contacts in different CDR sequences in both chains from each TCR as a function of *TRA* and *TRB* junction lengths (Supplementary Fig. 10). As expected, the *TRA* and *TRB* CDR3 regions contributed the most peptide contacts. There was a significant positive relationship between the

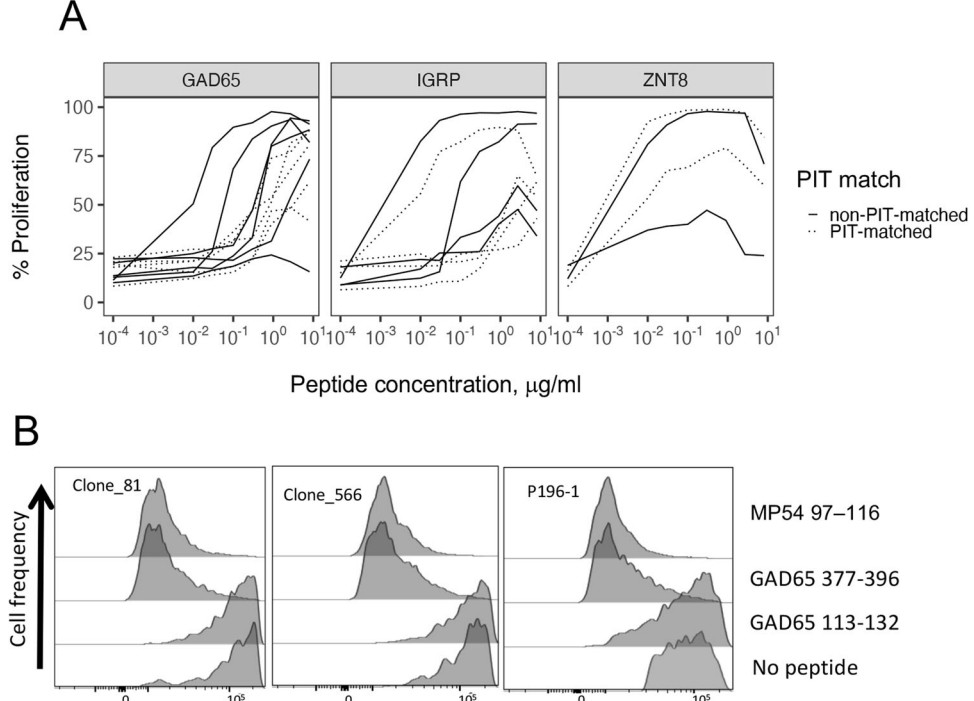

**Fig. 4 | Functional properties of selected PIT-matching and non-matching *TRA* chains of known specificity. A** Dose response curves for TCRs recognizing GAD65 (*n* = 11), IGRP (*n* = 8) and ZNT8 (*n* = 4) epitopes[12]. Curves represent single values for each point. Curves for multiple epitopes per target antigen were pooled to increase power; examination of curves for individual epitopes yielded conclusions consistent with the pooled data. **B** Cross-reactivity of TCRs with PIT-matching *TRA* junctions (Table 5) for autoimmune (GAD65) and viral (influenza M protein) epitopes in a CFSE proliferation assay.

number of peptide contacts mapping to the *TRA* CDR3 region and *TRA* junction length, and a weaker negative relationship with the *TRB* CDR3 region (Supplementary Fig. 10). While the number of contacts was low, there was also a negative relationship between the *TRA* CDR1 region and *TRA* junction length.

### Increased dependence on germline-encoded residues for binding of shorter and PIT-matching TCRs

The variation in contact residues by junction length (and PIT match) may indicate increased overall numbers of peptide contacts by longer TCRs, whereby increased numbers of CDR3 contacts would augment contacts CDR1 and CDR2. Alternatively, decreased numbers of CDR3 contacts in shorter TCRs could come with increased contacts in other regions. To distinguish these possibilities, we compared the overall number of contacts by TCR chain in IAR PIT-matched versus PIT-non-matched TCRs. This revealed no significant difference in overall peptide contacts in either *TRA* or *TRB* chains between the two groups (Fig. 3C). The ratio of *TRA* CDR1 to CDR3 contacts varied inversely and significantly with *TRA* junction length (p-value < 1e-4 for line slope, by linear modeling), while maintaining the separation in length between PIT-matched and PIT-non-matched junctions (Fig. 3D). This indicates that peptide recognition by shorter PIT-matched TCRs shows greater reliance on the germline-encoded residues in the *TRA* CDR1 region. Longer PIT-non-matched TCRs, in turn, show greater reliance on CDR3 residues generated by V(D)J recombination.

### IAR CD4 + T cell TCRs with PIT-matching *TRA* junctions show evidence of multi-specificity

We hypothesized that different peptide binding modes of PIT-matched and non-matched *TRA* junctions would lead to altered peptide recognition properties, such as strength and/or specificity of binding. As a measure of binding strength, we compared the functional avidities of TCRs having PIT-matched and non-matched *TRA* junctions specific for different islet peptide epitopes[12]. We plotted cell proliferation[12], versus peptide concentration (Fig. 4A) for the subset of PIT-matched or non-matched *TRA* junctions with known specificity. To increase power, we aggregated results with different peptides from each of three different islet antigens (GAD65, IGRP and ZNT8). These results showed wide variability in range of dose responses, but no consistent differences between TCRs with PIT-matched and non-matched *TRA* junctions. EC50 values[12] for aggregated GAD65- and IGRP-specific TCRs with PIT-matched and non-matched *TRA* junctions did not differ significantly by unpaired Wilcox tests; since there were only two ZNT8 TCRs, a p-value could not be calculated. We obtained similar overall results when individual peptides were considered. These data suggest that there are not large differences in functional avidity between TCRs with PIT-matched and non-matched *TRA* junctions.

Based on these results, we next hypothesized that TCRs with PIT-matched and non-matched *TRA* junctions differed in peptide binding specificity. This was supported by our previous finding of multi-specificity of some public IAR TCRs[12]. Though limited in number (*n* = 3), the frequency of these multi-specific clones was higher in public compared with private clones[12]. In the present studies, we noted that *TRA* chains from all three of these multi-specific TCRs were PIT-matching. This suggested that multi-specific *TRA* junctions were modestly more frequent in PIT-matching than non-matching *TRA* junctions (one-sided p-value = 0.030, Fisher's exact test).

Two of these PIT-matching multi-specific TCRs (Clones 81 and 566) recognized non-overlapping GAD65 epitopes[12]. In parallel and independent experiments, we unexpectedly found that the *TRA* chain of a TCR (P196-1) from Influenza A/MP54- reactive CD4 + T cells (Methods) perfectly matched the *TRA* chain from Clones 81 and 566 (Table 5). Furthermore, P196-1 had *TRB* chains that differed from Clones 81 and 566 at only a single AA position (Table 5). This sequence similarity led us to reason that closely related Clone 81, Clone 566 and P196-1 were all multi-specific. As a test, we compared the ability of

**Table 5 | Mismatched *TRB* chains from multi-specific TCRs with perfectly matched *TRA* chains. ᴬ**

| Clone ID | Specificity | Epitope | *TRBV*-gene | Junction | *TRBJ*-gene |
|---|---|---|---|---|---|
| Clone_81 | Islet | GAD65 377-396 | TRBV12-4 | CASS**P**QGGNTEAFF | TRBJ1-1 |
| Clone_566 | Islet | GAD65 377-396 | TRBV12-4 | CASS**V**QGGNTEAFF | TRBJ1-1 |
| P196-1 | Influenza | MP54 97–116 | TRBV12-4 | CASS**L**QGGNTEAFF | TRBJ1-1 |

ᴬShown are mismatched *TRB* chain amino acid sequence comparisons of multi-specific TCRs that recognized both GAD65 and MP54 peptides. All TCRs had identical *TRA* chains (TRAV29DV5-CAASRYSGGGADGLTF-TRAJ45). Sequences of GAD65 377-396[67] (H**KWKLSGVERAN**SVTWNPHK, where bold font denotes core sequence complexed with the TCR) and MP74 97-116[68] (VKLYR**KLKREITFHGA**KEIS) peptides showed minimal sequence similarity by multiple sequence alignment.
ᴮBold font indicates mismatches between the three *TRB* chains.

activating peptides (MP54 97-116 and GAD65 377-396), a non-activating peptide (GAD65 113–132) to trigger proliferation of TCR-transduced primary CD4 + T cells (Fig. 4B). We found that both MP54 97-116 and GAD65 377−396 peptides identically activated Clone 81, Clone 566, and P196-1 TCRs. Thus, these TCRs with PIT-matched *TRA* junctions were all multi-specific, supporting our hypothesis that distinctive TCR sequence features, including shorter and more hydrophobic *TRA* junctions, are linked to an inherent tendency for multi-specificity.

To elucidate possible mechanisms of cross-reactivity of the Clone 81 TCR for the MP54 97-116 and GAD65 377-396 peptides, we constructed molecular models[36] of the Clone 81 TCR[12] and complexes of these peptides together with HLA-DRA1*0101/DRB1*0401 molecules[37]. The cognate GAD65 377-396 (H**KWKLSGVERAN**SVTWNPHK, where bold font denotes core sequence complexed with the TCR) and MP74 97-116 (VKLYR**KLKREITFHGA**KEIS) peptides did not show compelling sequence similarity by multiple sequence alignment. They also showed no evidence of structure[38] or aromatic side chain[39] conservation characterizing "hotspots" of molecular mimicry used by some cross-reactive TCRs. Despite their low degree of peptide sequence similarity, molecular models of Clone 81 TCR with both peptides yielded identical scores of 0.88. In addition, *TRA* and *TRB* chains in both models showed similar predicted topography in their interactions with peptide-MHC class II complexes (Supplementary Fig. 11A, B). Overlapping but non-identical residues in the modeled TCR *TRA* and *TRB* chain CDR1, CDR2 and CDR3 regions contacted the different peptides (Supplementary Fig. 11C). Alignment of predicted structures of the *TRA* (Supplementary Fig. 11D) and *TRB* (Supplementary Fig. 11E) chains from models made with the different peptides showed nearly perfect superposition except in the CDR3 regions. This suggests that the PIT-matched Clone 81 TCR accommodates quite different peptide sequences through interactions involving conformationally conserved CDR1 (and CDR2 chains for *TRB*) regions, together with more variable CDR3 regions.

**Multi-specific TCRs shared sequence features with PIT-matched TCRs**

We wished to test in greater depth our hypothesis that PIT-matched and non-matched *TRA* junctions differed in peptide binding specificity. Recognizing that the number of known multi-specific TCRs from IAR T cells was too small at present to enable firm population-based conclusions about multi-specificity, we took an alternative approach to test our hypothesis. In other experiments, we observed that *VDJdb*, a curated public database of TCRs with known specificities[40], contains TCR sequences that recognize both single and multiple peptide epitopes. These designations were likely biased, with some TCRs being designated as single specificity because they were understudied, and others being designated as multi-specific based on reactivity with closely related peptides. We reasoned, however that these errors would tend to offset each other, and that the size and scope of this database would provide a more comprehensive source of specific and multi-specific TCRs, which could be used to test their junction sequence features. There were $n = 17,826$ unique *TRA* chains in *VDJdb* that recognized single epitopes and $n = 1664$ *TRA* chains that recognized multiple epitopes (Fig. 5A). We found that *TRA* CDR3 regions

(Fig. 5B) from TCRs that recognized multiple epitopes were significantly shorter (by ~3 nt in median length, or ~1 AA) than junctions from TCRs that recognized single epitopes. Likewise, *TRA* chains from multi-specific TCRs contained fewer non-templated (N region) nucleotides (~1 nt in median length) than TCRs with single specificity (Fig. 5C). Finally, *TRA* junction amino acid sequences from multi-specific TCRs were more hydrophobic in TCRs that recognized multiple epitopes (Fig. 5D). None of these differences were seen with *TRB* chains (Supplementary Fig. 12A–D). Thus, multi-specific TCRs from *VDJdb* shared multiple sequence features with PIT-matched TCRs, including shorter *TRA* but not *TRB* chains, had fewer N region nt and encoded more hydrophobic AA sequences.

## Discussion

It has long been unclear whether and how rare autoreactive cells in peripheral blood represent autoimmunity in target organs. In other words, are these cells drivers or passengers in autoimmune processes? To help resolve this question, we show here that IAR CD4 + T cell TCRs from peripheral blood share matching *TRA* chains, and lesser numbers of *TRB* chains, with PIT TCRs. We also show suggestive, although non-conclusive evidence that frequencies of PIT-matching *TRA* chains in blood increase prior to the time of diagnosis. This may indicate a temporal linkage of levels of PIT-matched *TRA* chains in blood with disease progression. Shared *TRB* chains between peripheral blood and insulin-reactive TCRs from the pancreas were reported recently[19]. These combined studies place potentially autoreactive public TCR chains at the scene of disease (the pancreas) at the right time (early in the disease process) to influence disease development.

There are several potential functional roles for these shared, public *TRA* junctions. Since PIT-matched TCRs, especially *TRA* junctions, show TCR convergence, it is possible that some PIT-matched TCRs share islet autoantigen binding. Convergence in biology is often associated with selection, and convergence of TCR sequences has been associated with antigen specificity in other systems[28–30]. Direct evidence to support shared specificity in the present case is lacking, largely because it is unknown how many PIT-matched IAR *TRA* chains show functional autoreactivity. It is not yet possible to accurately predict that TCRs share specificity without complete identity of both chains, necessitating laborious experimental work to determine TCR specificity.

It is also possible that the sharing of *TRA* and divergence of *TRB* junctions reflects different roles of these chains in TCR binding. For example, others have shown that the *TRA* chain in a "*TRA* centric" MHC class I-restricted TCR determines antigen specificity, with paired *TRB* chains regulating avidity[41]. Finally, it is possible that there has been selection for another function of PIT-matched TCRs other than directly recognizing islet antigens. Perhaps sharing and divergence arise from selection in the thymus. Yet another possibility, suggested by enrichment of TRAV41*01 *V genes* in PIT-matched TCRs, is that sharing and convergence reflect evolutionary remnants of a system of innate-like T cells in mammals[35]. Although out of the scope of the present study, it is also worth considering whether PIT-matched *TRA* junctions are related to dual *TRA* chain TCRs that have been proposed to modify various aspects of T cell function, including thymic selection[42].

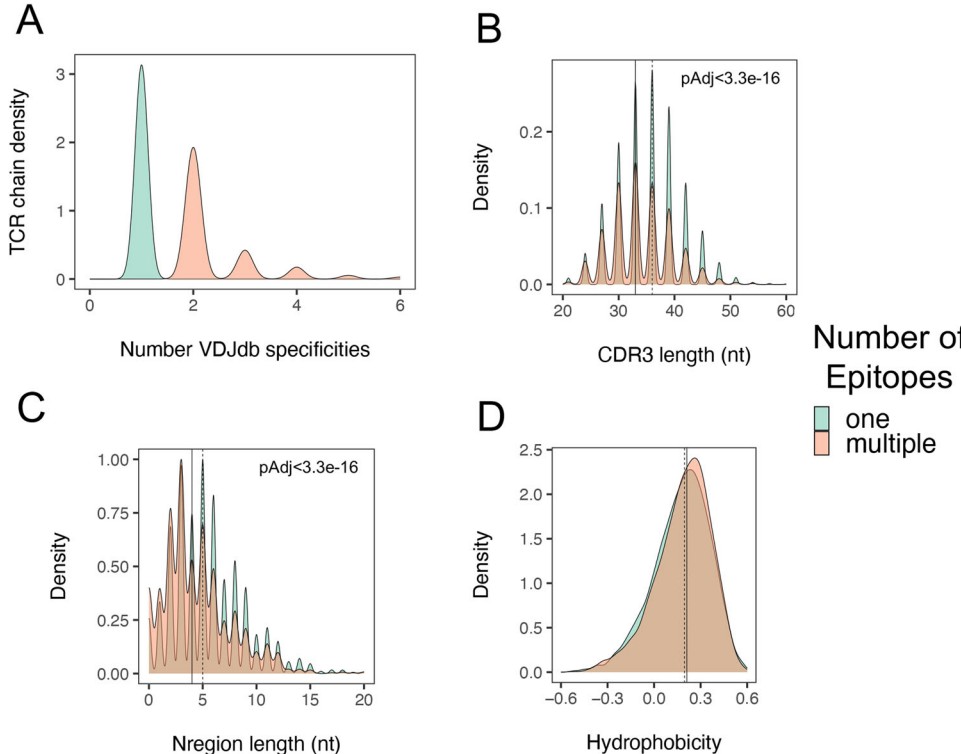

**Fig. 5 | *TRA* junctions from cross-reactive TCRs in the *VDJdb* database shared junction sequence features with PIT-matched *TRA* junctions. A** Density of *TRA* chains from TCRs with one versus multiple specificities in *VDJdb*. There were *n* = 17,826 unique *TRA* chains from TCRs with single specificity and *n* = 1664 with multiple specificities. **B** Distribution of *TRA* CDR3 nt lengths from TCRs with one versus multiple specificities; **C** Distribution of *TRA* N region nt lengths. **D** Distribution of *TRA* junction AA hydrophobicity. Solid vertical lines, median values from *TRA* chains from multi-specific TCRs; dashed vertical lines, median values from TCRs with single reported specificities. Significance of differences in distributions were assessed using two-sided Kolmogorov-Smirnov tests.

T cell specificity is key to cellular immunity[43]. Paradoxically, TCR cross-reactivity is necessary because the diversity of TCRs, while huge, is dwarfed by the vast array of potential foreign peptide-MHC complexes[44,45]. Cross-reactivity expands the potential range of a given repertoire towards foreign antigens, but it comes at the expense of greater potential for increased self-reactivity or autoimmunity. These considerations emphasize the importance of understanding of sequence and structural features determining TCR cross-reactivity. Unfortunately, at present such an understanding remains incomplete[43,46]. Our data show that cross-reactive (multi-specific) TCRs share sequence features with PIT-matched TCRs, namely shorter and more hydrophobic *TRA* junctions and diverse *TRB* chains, reminiscent of a previous report in mouse[47]. Supporting this conclusion, the few islet-specific TCRs for which we have demonstrated cross-reactivity are PIT-matched TCRs. Earlier work demonstrated shortened[32] and more hydrophobic[31] *TRB* chains in bulk T cell subsets from T1D donors, and suggested that these features are important in development of self-reactive T cells.

Molecular modeling showed that germline-encoded CDR1, rather than CDR3, residues from TRAV12-2 contributed critically to the binding of an immunodominant TCR to the Yellow fever virus[48]. This feature may contribute to the high precursor frequency and immunodominance of this TCR[48]. PIT-matching IAR T cell *TRA* chains also employed less non-templated and more germline-like (innate) mode(s) of epitope engagement. *TRA* CDR3 regions of PIT-matching IAR varied from the genome by relatively few N region nucleotide sequences. In addition, PIT-matching IAR T cell *TRA* chains showed shorter CDR3 regions and longer *TRAJ* segments than non-matching IAR T cell *TRA* chains. TCR CDR1 regions contact both peptide epitope and MHC residues, while CDR2 regions generally contact only MHC residues[23]. Thus, TCR regions important for peptide binding were shortened in

PIT-matching *TRA* chains and had altered peptide binding properties. Our results also suggest that these shorter and more hydrophobic, germline-like, *TRA* chains are a feature of many cross-reactive TCRs. While some TCRs that predominantly utilize germline-like *TRA* CDR3 regions are cross-reactive[49], this has not been previously demonstrated on the global level we show here.

One caveat to our study is that we have addressed only sharing of public TCRs. This was by necessity, since TCR sequences from IAR and PIT T cells from the same individuals were not available. While this feature enhances broadness of potential translational applications, it also limits mechanistic conclusions. Future studies may be able to address this weakness by testing of additional nPOD tissues[18], including spleen and pancreas draining lymph nodes, from the same donors used for PIT TCR identification. Another limitation is that our study is underpowered with respect to numbers of donors and for TCRs of known specificity needed to directly test our hypothesis that PIT-matched TCRs tend to be multi-specific. It also is important to note that the TCR repertoires we compared were all determined using different technologies, which can lead to systematic biases[50]. A more comprehensive collection of autoreactive TCRs with known specificity from more donors, and matched comparator repertoires determined using the same technology, remain aspirational goals for future studies.

Our studies have potential translational implications. Typically, studies on TCRs as biomarkers have taken a reductionist approach by focusing on one or a few TCRs to monitor disease progression and/or response to therapy[19,51,52]. Our broader-based studies place TCRs with distinctive *TRA* sequence features and potential cross reactivity in both the pancreas and blood in early stages of T1D progression. These distinctive *TRA*-centric TCRs may provide an alternative source of potential biomarkers and targets for future translational studies.

## Methods

### Donors and samples

Donor group characteristics are summarized in Supplementary Data 1. The donor groups were defined as follows: HC, age-matched healthy donors; 1 Aab, donors at risk for T1D having one islet-directed auto-antibody; 2Aab, at risk donors having two or more islet-directed autoantibodies; newT1D, T1D donors, within three months of original diagnosis; and T1D, donors with established T1D for a mean of 3.79 years (range, 1.6 −7.1 years). IAR CD4 + T cells were isolated from peripheral blood or cryopreserved PBMC samples for these studies (Supplementary Data 1).

*Cohort 1* samples from HC, newT1D, and T1D participants were described previously[12]. Briefly, T1D donors were from the Benaroya Research Institute Disease Registry and Repository (BRI DRR). newT1D samples were from the placebo arms of the AbATE (NCT00129259) and START (NCT00515099) clinical trials sponsored by the Immune Tolerance Network[53,54]. HC samples were collected following informed written consent from healthy volunteers, matched for age and sex to T1D individuals with T1D and had no personal or family history of T1D. HC, T1D, and most newT1D donors had high-risk DRB1*0401 HLA class II alleles. Participant characteristics were summarized previously[12]. Samples from the BRI DRR were collected after informed written consent from donors, with approval from the BRI Institutional Review Board (IRB7109-332).

*Cohort 2* samples were from the Type 1 Diabetes TrialNet Pathways to Prevention study (TN01) for the at-risk donors. Samples from TN01 participants were collected under the auspices of TrialNet ancillary study #201 and analyzed under ancillary study #200. Samples from TN01 donors were collected after informed written consent by the BRI Immune-mediated Disease Registry and Repository for sample storage and distribution (IRB07109).

Pancreas tissue or purified islet cell samples were obtained from organ donors identified by the Network for Pancreatic Organ Donors with Diabetes program (nPOD; RRID:SCR_014641[18], a protocol at Vanderbilt University Medical Center/University of Pittsburgh[16]; the Integrated Islet Distribution program (IIDP; RRID:SCR_014387); and the Alberta Diabetes Institute Islet Core (ADI). Organ Procurement Organizations (OPO) partnering with nPOD to provide research resources are listed at https://npod.org/for-partners/npod-partners/. nPOD obtains pancreas and other tissues from deidentified, deceased organ donors from partnering organ procurement organizations (OPOs) in the US under IRB Protocol #201600029 from the University of Florida. OPOs obtain consent for research from the donors' families. The use of sequencing data from cadaver spleen and pancreas nPOD samples was approved by the BRI IRB as presenting no more than minimal risk to human donors and, therefore, qualifying for a Certificate of Exemption under 45 CFR 46.104(d)(4)(ii). Information, which may include information about biospecimens, was recorded by the investigator in such a manner that the identity of the human subjects cannot readily be ascertained directly or through identifiers linked to the subjects.

### IAR CD4 + T cell TCRs

We isolated IAR T cells from peripheral blood using an overnight activation induced marker (AIM) assay based on upregulation of CD154 and CD69[12,13]. Samples from Cohort 1 were stimulated with a peptide pool of immunodominant DRB1*0401, DRB1*0301, and DQ8 restricted peptides from the islet proteins GAD65, IGRP, ZnT8, and preproinsulin (20 aa in length)[12,13]. In short, PBMC were stimulated with a pool of 35 immuno-dominant peptides from GAD65, IGRP, ZnT8, and preproinsulin restricted to the high-risk HLA class II DRB1*0401, *0301, or DQ8 molecules[12,13]. PBMC from donors in Cohort 2 were stimulated in an HLA agnostic approach with overlapping peptide libraries (20 amino acids (AA) in length, 12 AA overlap) from the above islet proteins (Supplementary Data 4). For both groups, CD154+ islet peptide activated cells were magnetically enriched, isolated as CD154 + CD69+ by single cell flow sorting and subjected to scRNA-seq to identify paired *TRA* and *TRB*

chains in IAR T cells. The flow cytometry gating strategy used for Cohort 2 samples is shown in Supplementary Fig. 13 and antibodies are listed in Supplementary Data 5. Prior to use in this study, IAR CD4 + T cell TCRs with in-frame protein sequences were selected and filtered by removing iNKT cells with the CVVSDRGSTLGRLYF junction ($n = 21$ cells, 47 junctions); and MAIT cells with the TRAV1-2 *V gene* and TRAJ33, TRAJ20 or TRAJ12 *J genes* ($n = 18$ cells, 45 junctions). Complied and filtered TCR sequences are presented in Supplementary Data 2 and at https://www.ncbi.nlm.nih.gov/geo/query/acc.cgi?acc=GSE256481. Some sequences were described previously[12] and are also available at https://www.ncbi.nlm.nih.gov/geo/query/acc.cgi?acc=GSE182870

The P196-1 TCR was identified in experiments similar to those previously described for IAR CD4 + T cells[12], but using the Influenza A MP54 97–116 peptide instead of islet peptides. The near identity of the P196-1 TCR to IAR CD4+ TCRs was discovered by sequence matching against an internal database of experimentally determined antigen specific TCRs identified using scRNA-seq. A single codon-optimized DNA fragment flanked by Not1-Spe1 restrictions and encoding the *TRA* and *TRB* junctions of the P196-1 TCR, integrated into the 'TCR flex' pMP71 sequence upstream of the murine *Trac* and *Trbc* genes[55] (Supplementary Data 6), was prepared synthetically (GenScript, Piscataway, NJ). The synthetic fragment was then incorporated as a NotI and SpeI restriction fragment in place of the green fluorescent protein (GFP) sequence in the lentiviral vector, pRLL-MND-GFP[56].

### PIT TCR sequences

TCR alpha and beta chain sequences expressed by T cells in the islets or pancreas tissues were amplified by PCR[57,58]. Briefly, single cells stained with anti-CD3 and anti-CD4 or anti-CD8 antibodies were sorted into each well of 96-well plates, followed by reverse transcription synthesis of cDNA. TCR alpha and beta chain genes were amplified by multiplex PCR using primers binding to *V gene* and the constant region, Illumina sequencing linkers were added by additional PCR, and fragments were sequenced on Illumina sequencers. *V gene, J gene*, and junction sequences were identified using IMGT/HighV-QUEST[59]. TCRs having with in-frame protein sequences were selected for further analysis. IAR *TRA* junctions having 0 or 1 mismatches with PIT TCRs were designated as "PIT-matched". *TRB* junctions were considered "PIT-matched" if they were paired with a PIT-matched *TRA* junction.

### Other TCR sequences

TCR sequences from COVID-19 patients and HC[21] were obtained from https://www.ebi.ac.uk/biostudies/arrayexpress/studies/E-MTAB-9357/files. *VDJdb*[40] TCR sequences (versions 2022-11-24 and 23-09-16) were downloaded from https://vdjdb.cdr3.net/about. *VDJdb* V/J/CDR3 protein sequences from *VDJdb* (updated 09/16/23) were converted into complete coding sequences representing fully spliced TCR cDNA sequences using *Stitchr*[60]. TCR sequence features were derived from *Stitchr*-produced cDNA sequences by IMGT/HighV-QUEST[59]. In some cases, comparisons were made between groups of TCRs of different sizes. To confirm that this disparity between different set sizes did not bias the results, we normalized group sizes, down sampling by randomly selecting with replacement subsets of the larger group(s) matching in size with the smaller group. Comparisons were made between results obtained before and after down sampling to ensure that the comparison made with the full data set was accurate. We repeated these comparisons multiple times to ensure that the down sampling comparison was representative. Reported results utilized the full data sets.

### Transduction of human CD4 + T cells

Human CD4 + T cells were transduced with lentiviral TCR expression vectors[12]. Briefly, CD4 + T cells from HC donors were stimulated for with human anti-CD3/CD28 monoclonal antibodies (mAbs), then incubated with recombinant TCRs cloned into a modified 'TCR flex' pMP71 vector[12] upstream of the murine *Trac* and *Trbc* genes[55].

Transduction efficiency was determined by staining with a mAb targeting the murine *Trbc* chain (H57-597, allophycocyanin (APC)-labeled, BD Biosciences, San Jose, CA). Transduced T cells were labeled with 5(6)-Carboxyfluorescein diacetate N-succinimidyl ester (CFSE) and mixed with irradiated antigen-presenting cells (APC) loaded with antigenic peptides. APC were either autologous PBMC; PMBC from MHC class II-matched individuals; or Priess lymphoblastoid cells, (ATCC, Rockville, MD). After incubation at 37 °C, cells were stained with anti- human CD4+ (Clone RPA-T4, phycoerythrin-labeled, BioLegend, San Diego CA), and anti- murine *Trbc* chain to identify CFSE dye dilution in transduced cells by flow cytometry[12].

### Flow cytometry
Antibodies used for flow cytometry are described in Supplementary Data 5. Peptide simulated PBMC cultures were surface stained with anti-CD154-PE followed by anti-PE-magnetic microbeads (Miltenyi) according to the manufacturer's instructions. Magnetically enriched CD154+ cells were surfaced stained with PerCP-Cy5.5-coupled monoclonal antibodies specific for CD8, CD19, CD14, CD56, iNKT, and CD161 along with Viaprobe live/dead stain (BD, Franklin Lakes, NJ) as a dump channel. Monoclonal antibodies specific for the following surface markers were included to identify and phenotype IAR CD4 + T cells: CD3, CD4, CD45RO, CD45RA, CCR7, CD69, CCR4, CCR6, CXCR3, CD95, CD25, CD2, CD38, and SLAMF6. IAR CD4 T cells were identified as CD3 + CD4 + CD154 + CD69 + T cells according to the gating scheme shown in Supplementary Fig. 13 and single cell sorted into a 96 well plate using a BD FACSAria flow cytometer. Proliferation of CFSE-labeled TCR transduced CD4 T cells was determined by surface staining cultured cells with anti-CD4-PE and anti-murine Trbc-APC monoclonal antibodies (Supplementary Data 5) followed by flow cytometry using a BD LSRFortessa cytometer. CFSE intensity was assessed in CD4+mTrbc+ cells (Fig. 4B).

### TCR expansion and sharing
TCR expansion was defined as the number of individual cells expression the same junction AA sequence. A junction was designated as "expanded" if the number of cells with the same junction AA sequence was >1. Conversely, a junction was designated as "non-expanded" if it was found in only one cell. TCR publicity for each junction in a TCR was defined as the number of donors expressing the same junction AA sequence. A "public" junction was defined as one found in multiple donors; a "private" junction was expanded and found only in a single donor. We used *IGoR* software (v. 1.4.0) to estimate the generation probability (*Pgen*) of junction sequences during TCR recombination[26]. Junction sequences with higher (or less negative) *Pgen* values have a higher probability of generation by V(D)J recombination. TCR convergence was defined as the number of nucleotide (nt) sequences encoding each junction AA sequence. Junctions having >1 nt sequence associated with a single AA sequence were designated as "Converged" and junctions having only a single nt sequence per AA sequence, "non-Converged".

### Sequence comparisons
Pairwise Levenshtein distances between peptide sequences were calculated using the *stringdist* (v. 0.9.10) software package in R. The Levenshtein index is the number of residue changes needed to transform one sequence into another. Motif mapping using the *MEME* software was accomplished at https://meme-suite.org/meme/tools/meme. Sequence features of TCR chains were determined by IMGT/HighV-QUEST[61] analysis of nt sequences.

### Sequence features from *VDJdb* TCRs
Since nt sequences used to identify junction sequence features are not readily available for *VDJdb* TCRs, we utilized the software package, *Stitchr* (v. 1.1.3.1), to produce complete TCR cDNAs from V/J/CDR3 AA sequences[60]. We then used IMGT/HighV-QUEST (v. 1.9.4)[61] to identify TCR sequence features from the predicted cDNA sequences.

### TCR modeling
For modeling, we used TCRmodel2, which is based on the AI system, AlphaFold v2.3[62], a newly described method that predicts protein structure with high accuracy at unmatched scale[63]. TCRmodel2 uses focused databases of TCR and MHC sequences to expedite multiple sequence alignment feature building; optimization of the TCR template selection; and utilization of peptide–MHC complex structures as templates to improve modeling accuracy. We used the TCRmodel2[36] web server (https://github.com/piercelab/tcrmodel2) to predict TCR–pMHC complex structures. We saved best fit models for further analysis; these had Model Confidence scores[36] from 0.74–0.91 (0.85 ± 0.04, mean ± SD), where a score ≥ 0.85 is considered a "likely good" model. Models were viewed and analyzed using the Mol* Viewer[64] (https://molstar.org/viewer/). TCR-peptide contact residues, (i.e., TCR residues <5 Å in distance from the bound peptide chain) were identified and mapped to TCR sequence features identified using IMGT HighV-QUEST[59]. Structural alignment was performed using the RCSB PDB[65] web-based viewer (https://www.rcsb.org/alignment).

### Statistics
Statistical tests were performed using the R programming language and software environment (v. 4.2.3). We utilized t-tests for group comparisons of continuous, normally distributed variables; Wilcoxon signed rank tests for non-normally distributed variables; Kolmogorov–Smirnov tests for distributions of numeric values; the hypergeometric test to determine the significance of set overlaps; and Fisher's exact test for categorical variables. Unless otherwise noted, we performed non-paired, two-sided tests and assumed equivalent variation in the groups compared. Where appropriate, multiple testing corrections were made[66]. The term "significant" is reserved for p-values (single tests) or false-discovery rate (FDR)-adjusted p-values (pAdj) (multiple tests) of <0.05. Only significant P- and pAdj values are indicated on the Figures. Specific tests used to derive each listed P- or pAdj value are given in the text or in the Figure legends.

### Inclusion and ethics
Benaroya Research Institute is committed to creating an environment where all people are, and perceive themselves to be, welcomed, valued, respected, and heard in order to develop, contribute to and achieve aligned organizational and scientific goals.

### Study approval
Protocols for these studies were approved by the Institutional Review Board of Benaroya Research Institute (IRB7109-332 and IRB7109-460.02, for *Cohorts* 1 and 2, respectively). Protocols for collection of nPOD samples were approved under an IRB from the University of Florida (#201600029). Protocols for the clinical studies were approved under the auspices of NCT00129259 for the AbATE trial[53] and NCT00515099 for the START study[54]

### Reporting summary
Further information on research design is available in the Nature Portfolio Reporting Summary linked to this article.

## Data availability
The profiles yielding TCR data generated in this study from *Cohorts 1* and *2* have been deposited in the GEO repository under accession numbers GSE182870 and GSE256481. PIT TCR sequences were deposited in the iReceptor repository [https://gateway.ireceptor.org/samples?query_id=98188]. TCR sequences and models (pdb files) are also available from Figshare [https://doi.org/10.6084/m9.figshare.24309679]. All data files used to generate figures are available at GitHub [https://github.com/BenaroyaResearch/Germline-like-TCR-alpha-chains-shared-between-autoreactive-T-cells-in-blood-and-

pancreas]. All raw data are freely available without restrictions. Source data are provided with this paper.

## Code availability

Code and data for generating Figures, including TCR sequences and pdb files of molecular models, are available at https://github.com/BenaroyaResearch/Germline-like-TCR-alpha-chains-shared-between-autoreactive-T-cells-in-blood-and-pancreas.

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

## Acknowledgements

We acknowledge Vivian Gersuk, Kimberly O'Brien, Jeffry Yaplee, Thao Huynh, and Quynh-Anh Nguyen for molecular profiling assistance; the BRI TrialNet Clinical Center led by Dr. Carla Greenbaum and Dr. Sandra Lord for collecting samples from individuals with and at-risk for T1D; and Mario Rosasco for assistance with TCR analysis. This research was performed with the support from the Immune Tolerance Network (5UM1AI109565 to Gerald T Nepom), the American Diabetes Association (1-19-ICTS-006 to KC), and the Network for Pancreatic Organ donors with Diabetes (nPOD; RRID:SCR_014641), a collaborative type 1 diabetes research project supported by JDRF (nPOD: 5-SRA-2018-557-Q-R) and The Leona M. & Harry B. Helmsley Charitable Trust (Grant#2018PG-T1D053, G-2108-04793). The content and views expressed are the responsibility of the authors and do not necessarily reflect the official view of the ADA or nPOD. The Type 1 Diabetes TrialNet Study Group is a clinical trials network funded through a cooperative agreement by the National Institutes of Health (NIH) through the National Institute of Diabetes and Digestive and Kidney Diseases (NIDDK), the National Institute of Allergy and Infectious Diseases (NIAID), and the Eunice Kennedy Shriver National Institute of Child Health and Human Development, and JDRF.

## Author contributions

P.S.L., M.N., and A.P. conceived the analysis; P.S.L. and K.C. isolated and provided IAR T cell sequences; M.N. isolated the PIT TCR sequences in her laboratory and provided them for analysis; E.B., J.C., and F.B.W. performed laboratory experiments; E.S. and C.S. designed the cohorts and collected samples; P.S.L., T.B., and S.B. analyzed the TCR data; P.S.L., K.C., and M.N. interpreted data; P.S.L. wrote the manuscript, with contributions from all authors.

## Competing interests

The authors declare no competing interests.
