## [Peer Review File · Nature Communications]

Self-reactive germline-like TCR alpha chains shared between blood and pancreasREVIEWER COMMENTS

Reviewer #1 (expert in TCR sequencing analysis):

Authors describe TRA sequences observed in islet antigen reactive (IAR) cells, detectable also in T cells isolated from pancreatic infiltrations (PITs) and .

Starting point of the paper is an observation that some TRA chains of IAR cells are also observed in the pancreatic infiltrates of other individuals (irrespective of the disease status IAR and PIT donor). Then they compare various characteristics (length, non-templated edits etc.) of such PIT-overlapping TRAs versus PIT-nonoverlapping TRAs.

They also infer whether the PIT-matched TRAs TCRs are multi-specific, by comparing their characteristics to the characteristics of known multi-specific TRAs.

Autoimmunity leading to T1D development is a very difficult area, as the targeted individuals and tissues are not easily accessible. Hence any new data in this area is of great interest. However, many conclusions are overinterpretation of the data and the abstract contains several statements not supported by the presented data.

1. The major issue for me is treating the same TRAs as proxy for the same TCR specificity. However, even the identical TRA sequence does not mean necessarily the same TCR specificity, as both TCR chains and the interacting HLA molecule decide about interaction between T cell and antigen. In searching for overlaps between TRAs repertoires authors do not control for HLA genes or alleles, even for their interaction with HLA-I or HLA-II (which differ greatly in the mechanics of antigen presentation). Thus, the premise of the same specificity of PIT cells and IAR cells, based on the identity of TRA chains is too strong and
The abstract statement "Using TCR sequences as barcodes, we measured infiltration of IAR T cells from blood into pancreas of organ donors with and without T1D" is a strong overstatement. In their previous work on IAR cells, the authors corroborated activation by islet antigen epitopes for only ~60% of tested IAR-isolated TCRs, meaning that even a perfectly matched cell (by both TCR chains, with controlled HLA) present in IAR and PIT sets is reactive to islet antigens. The authors discuss extensively the reasons for ~40% false positive rate in their assay in their JCI insight paper (ref 13), however in the current paper they treat all IAR cells as bona fide IA-specific.

2. Based on an observation that the overlap of IAR TRAs and PIT TRAs is higher than between any of two public datasets and PIT TRAs, authors conclude that IAR cells are selected towards PIT-like sequences.

Overlap of IAR TRA repertoires with those of PIT samples might be driven by HLA bias, not necessarily antigen-driven selection. HLA of IAR group is biased towards diabetes-high risk alleles (described by the authors); the same might be suspected for PIT individuals, dominated (8K out of 9.8K cells) by individuals with T1D autoimmunity detected by antibodies/T1D. Authors compare the overlap of these two datasets with overlaps of PIT/IAR with public datasets from COVID patients or healthy individuals. The latter groups are most probably non-HLA biased. HLA influence on sharing TCR sequences is a well known phenomenon and such comparison should control for it (for example use samples of non-selected repertoires from the same individuals as IAR were obtained or to matched samples in comparisons by HLA).

Also, the healthy/COVID data was obtained with different technology than IAR/PIT data, with inherently different sequence biases, this should be also taken into account when discussing explanations for reduced overlap. Because of these two issues, the statement that PIT-overlapping sequences are selected for reactivity to IA is contentious.

3. The other avenue not explored by the authors is that the PIT-overlapping sequences might be a subset of IARs which is public and all their characteristics would be related to their public status, not to their IA reactivity.

By definition and experimental design, the TRAs shared between individuals - between IARs and PITs - are public sequences. Diverse metrics of PIT-overlapping TRAs described by authors might be universal characteristics of public sequences, not necessarily antigen-selection related. This is hinted at by PIT-matching sequences being shorter and closer to the germline sequences (line 341), suggesting that they might be easier to generate by VDJ recombination and hence be

present in multiple individuals.

Besides of checking probability of generation of these TRAs, a good baseline would be overlap of non-antigen selected, TRA repertoires (and characteristics of overlapping versus nonoverlapping sequences) from individuals with matched HLA.

Also, the authors published previously on public sequences within IAR cells – comparison of the sequences identified in this work with the previously described would be of interest.

4. I also could not find any supporting data within the main text nor figures for the abstract statement:

"We detected extensive TCR sharing between IAR T cells from peripheral blood and pancreatic infiltrating T cells (PIT), with perfectly matched or single mismatched TRA junctions and J gene regions, comprising ~34% of unique IAR TCRs."

Table 1 lists all perfectly TRA-matched sequences: there are 10 of them (in 47/1,606 cells, number of unique IAR TRAs not provided) - it is unclear where the number 34% comes from, especially as authors do not state what is the number of 0/1-mismatch sequences.

Methods section is very scant and actually in many instances provides fewer details than the main text – e.g. the main text mentions magnetic separation of T cells, nowhere alluded to in the methods). Moving relevant details (like specific software used for string extraction/comparison) to the methods would make the paper more readable.

The paper would also benefit from shortening some parts, for example the reader does not need to know the path which lead the authors from comparison of TRA length to the comparison of V gene usage. A statement that TRA repertoires differ in their V gene usage and hence (because of V gene lengths) in TRA lengths would be easier to read.

In summary, even though the dataset is unique and important for the field, I think this work requires major revision, more cautious data interpretation and a rewrite before publishing.

Reviewer #2 (expert in type-1 diabetes immunopathogenesis):

Type 1 Diabetes is an organ-specific autoimmune disease, where many studies have shown that autoantigen-specific CD4 and CD8 T cells are involved. These cells recognize autoantigens through their T cell receptors (TCR) and, as such, understanding the nature of these TCRs is key to unravel the function of T cell in the pathogeny of the disease. Due to the complexities of sampling human pancreas, most studies have studied TCR from blood samples instead. Here, the authors go a step forward by analyzing TCR repertoires also in samples from human pancreas, finding that many TCRA chains shared by autoantigen-specific blood T cells and pancreas are of shorter length, more hydrophobic and potentially cross- reactive. While this approach is novel, the methodologies are appropriate, and the conclusions interesting, several issues remain:

Major concerns:

- Page 5, lines 109-121: here the authors explain the sequencing of pancreatic infiltrating T cells (PIT), but there is no information about numbers of donors, demographic information, etc. The authors should include this information in the manuscript, as they did for other study participants.
- Page 5, lines 124-125: here the authors explain how they did the matching of TCR sequences between Islet-antigen reactive CD4+ memory T cell (IAR) TCRs and PIT TCRs and indicate that sequence comparisons were made only at the amino acid level. To my opinion, this should be done also at the nucleotide level. Although this type of sharing is, in theory, less probable than perfect matches at amino acid level, the sharing of a TCR at the nucleotide level would be highly relevant, indicating identical thymic events between/among donors. Therefore, the analysis at the nucleotide level would add another level of significance to these results.
- Page 5, lines 128-129, and Page 7, line 172: although it becomes apparent that matching was higher for TRA, there was also matching for TRB. However, the authors only go on analysing TRA features- although understandable due to the higher effect observed, the fact of finding TRB sequences matching between IAR and PIT TCRs is an important finding and these matching TRB sequences should be further studied and described in the manuscript.
- Page 5, lines 129-131: if the referee understands this correctly, IAR cells are CD4+, while PIT

cells where CD4 or CD8. If this is correct, how is it possible that many PIT CD8 TCRA sequences were found in IAR, CD4+, repertoires?

- Page 5, lines 132-133: when comparing TCRs from IAR and PIT, one would assume that if a given absolute number of IAR TCRs appear in PIT repertoires, the same absolute number of PIT TCRs would appear in IAR ones. However, these numbers are different in the text (47 vs 44). Could the authors clarify?
- In general, the figures (main and supplementary) need to be improved for clarity, replacing "true" and "false" wording by the actual variable/group being shown (e.g. PIT-matching, non PIT-matching). This would ease the interpretation of the figures by the readers.
- Pages 9-10, lines 248-252: PIT matched TRA junctions were shorter and more hydrophobic, while their paired TRB were not. What about the PIT matched TRB junctions?
- Figure 3D: here the authors state that "Ratios of TRA CDR1 to CDR3 peptide contacts decreased significantly with increasing TRA junction length". However, one could argue that, at longer lengths of CDR3, there are higher probabilities of having higher number of contacts, in which case both variables are related and it would be obvious that the ratio CDR1 contacts/CDR3 contacts would be lower if CDR3 length is higher. The authors should better explain what the novelty of this statement is.
- Page 16, lines 445-447: "To help resolve this question, we show here that a significant fraction of IAR TCRs from peripheral blood share matching TRA chains with PIT TCRs, and vice versa". The fraction of sharing is of 0.45% for PIT in IAR, and 2.9% for AIR in PIT (figures from Page 5, lines 132-133) which, although relevant and important, is considered not significant. The authors should tone down this claim.
- Throughout the manuscript, on many occasions the authors mention results that are not shown. Given the importance of such statements for the overall message of the study, the following results should be shown/discussed, either as main or supplementary figures/tables, depending on space constraints and importance/relevance of the claims:
 - o Page 5, lines 133-135: "The distribution of perfect matches between different subject groups (HC, T1D and newT1D) and cell types (CD4+, 135 CD8+) did not differ significantly from the distribution of total TCR populations from each group".
 - o Page 8, lines 200-203: "We obtained results in Figures 1A-1E from Cohort 1 (Materials and Methods). To validate these findings, we repeated these analyses with all samples in an independent cohort (Cohort 2). We obtained essentially identical results, demonstrating that our observations were not restricted to a single data set and therefore had a potentially broader range of islet specificities". This is particularly important- results for this cohort should be shown, instead of simply mentioning that they are "identical".
 - o Page 8, lines 210-212: "This preliminary analysis revealed that the fraction of PIT TRA junction matches with expanded IAR T cells was significantly elevated, and the fraction of PIT TRA junction non-matches reduced, in newT1D subjects, relative to HC and T1D subjects".
 - o Page 9, lines 239-242: "We found significant overlap between public IAR T cell and PIT-matched TRA junctions (p-value = 1.46e-14, hypergeometric distribution) but not private IAR T cell and PIT-matched TRA junctions (p-value >0.05). Thus, there was strong overlap between cell populations with public and PIT-matching TCRs".
 - o Page 11, lines 285-287: "In other analyses, we found that framework (FR) regions FR1, FR2, FR3 and FR4 did not differ in length between PIT-matched and non-matched TRA chains, showing selectivity of the differences in CDR1 and CDR3 lengths".

Minor concerns and others:

- Introduction: it should be mentioned that blood and pancreas samples are not from the same donors, for clarity.
- Line 116: "represented 4,706 T cells". Do the authors mean "4,706 CD4+ T cells"?
- Page 6, line 162: if here authors refer with "HC" to those donors from Su et al, it should be made clearer for the readers, as they could confuse these HC with those sequenced in the current study.
- Page 8, line 209: has the size of down sampling been described in the methods section?
- Page 8, line 220: the sentence "in cells with expanded TCRs" is not clear, as a single cell can not have expanded TCRs. Do the authors mean expanded clones? If so, please rephrase for clarity.
- Figure 2, legend: it should explain what the brown histograms represent.
- Pages 11-12, lines 300-311: this paragraph could be moved after line 282, as it would ease the understanding of these section of the results.
- Pages 13-14, lines 365-366: do the authors mean "non-matching" or "single mismatching"?

- Page 14, line 379: the sequence of these multi-specific TCR clones is not shown; given the importance of these results, they should be shown, for example in Table 3.
- Page 14, lines 389-390: "TCR transduced primary CD4+ T cells". The methods for the generation of these cells are not shown.
- Page 15, line 414: it should read "greater".
- Page 18, line 500: it should read "utilize".
- Supplementary file, Page 3, lines 38-40: "Cohort 2 samples were stimulated with overlapping peptide libraries from the same islet proteins optimized for HLA class II presentation (20 aa, 12 aa overlap, Table S3)". The table does not seem to be included.

RESPONSE TO REVIEWERS' COMMENTS

We thank the reviewers for their thorough and positive reviews. We have revised the manuscript accordingly addressing each comment, paying particular attention to providing further evidence to support our conclusions.

Reviewer #1 (expert in TCR sequencing analysis):

Authors describe TRA sequences observed in islet antigen reactive (IAR) cells, detectable also in T cells isolated from pancreatic infiltrations (PITs) and .[sic]

Starting point of the paper is an observation that some TRA chains of IAR cells are also observed in the pancreatic infiltrates of other individuals (irrespective of the disease status IAR and PIT donor). Then they compare various characteristics (length, non-templated edits etc.) of such PIT-overlapping TRAs versus PIT-nonoverlapping TRAs.

They also infer whether the PIT-matched TRAs TCRs are multi-specific, by comparing their characteristics to the characteristics of known multi-specific TRAs.

Autoimmunity leading to T1D development is a very difficult area, as the targeted individuals and tissues are not easily accessible. Hence any new data in this area is of great interest.

1. However, many conclusions are overinterpretation of the data and the abstract contains several statements not supported by the presented data.

We apologize for any overinterpretation of our data. In the revised manuscript, we have presented more data to support our conclusions and endeavored throughout to make our wording consistent with the data shown. Please see below for several specific examples.

2. The major issue for me is treating the same TRAs as proxy for the same TCR specificity.

We agree with this point and never intended to treat these parameters as equivalent. We have modified our language throughout to make clear that TRA sequence identity does not necessarily equate to TCR specificity. We have also modified the revised Discussion to emphasize this point (lines 463-471).

However, even the identical TRA sequence does not mean necessarily the same TCR specificity, as both TCR chains and the interacting HLA molecule decide about interaction between T cell and antigen. In searching for overlaps between TRAs repertoires authors do not control for HLA genes or alleles, even for their interaction with HLA I or HLA II (which differ greatly in the mechanics of antigen presentation). Thus, the premise of the same specificity of PIT cells and IAR cells, based on the identity of TRA chains is too strong and [sic]

*We have added new data on TCR chain, cell type and HLA class II allotypes on the frequency of PIT matches (new **Figures 1A-C, H, and I, Figure S1 and Figure S2 A-C**). These data show that neither cell type nor HLA class II allele has a major effect on the frequency of PIT matches. With respect to cell type, we have added text and a reference showing that CD4+ and CD8+ T cells share a non-trivial fraction of TCR chains despite the difference in the mechanics of antigen presented by HLA class I and II (lines 115-117). We agree that the premise of the same specificity of PIT cells and IAR cells, based*

on the identity of TRA chains is too strong and have avoided making this conclusion throughout.

3. The abstract statement “Using TCR sequences as barcodes, we measured infiltration of IAR T cells from blood into pancreas of organ donors with and without T1D” is a strong overstatement.

We have changed the wording in the abstract to read, “We identified paired alpha/beta (TRA/TRB) T cell receptors (TCRs) in IAR T cells from the blood of healthy, at-risk, new onset, and established T1D donors, and measured sequence overlap with TCRs in pancreata from organ donors”. We believe this sentence better adheres to the data presented (lines 24-27).

4. In their previous work on IAR cells, the authors corroborated activation by islet antigen epitopes for only ~60% of tested IAR-isolated TCRs, meaning that even a perfectly matched cell (by both TCR chains, with controlled HLA) present in IAR and PIT sets is reactive to islet antigens. The authors discuss extensively the reasons for ~40% false positive rate in their assay in their JCI insight paper (ref 13), however in the current paper they treat all IAR cells as bona fide IA-specific.

We have added the following sentence to the revised text:

“Reason(s) why the remaining 18 TCRs tested did not demonstrate peptide specificity remain unknown but may involve suboptimal avidity, and/or presentation by MHC class II molecules not tested.” (lines 100-102).

5. Based on an observation that the overlap of IAR TRAs and PIT TRAs is higher than between any of two public datasets and PIT TRAs, authors conclude that IAR cells are selected towards PIT-like sequences. Overlap of IAR TRA repertoires with those of PIT samples might be driven by HLA bias, not necessarily antigen-driven selection. HLA of IAR group is biased towards diabetes-high risk alleles (described by the authors); the same might be suspected for PIT individuals, dominated (8K out of 9.8K cells) by individuals with T1D autoimmunity detected by antibodies/T1D. Authors compare the overlap of these two datasets with overlaps of PIT/IAR with public datasets from COVID patients or healthy individuals. The latter groups are most probably non-HLA biased. HLA influence on sharing TCR sequences is a well known phenomenon and such comparison should control for it (for example use samples of non-selected repertoires from the same individuals as IAR were obtained or to matched samples in comparisons by HLA).

*In the revised manuscript, we have examined the effects of different HLA class II alleles on PIT matching (new **Figures 1H and I, Figure S1D**). In our data, different donor HLA class II alleles do not have a major effect on the frequency of PIT matches (lines 204-205). We have added a statement with a reference about HLA-independent associations with TRB chains (lines 205-206).*

6. Also, the healthy/COVID data was obtained with different technology than IAR/PIT data, with inherently different sequence biases, this should be also taken into account when discussing explanations for reduced overlap. Because of these two issues, the statement that PIT-overlapping sequences are selected for reactivity to IA is contentious.

We acknowledge the reviewer’s point. Unfortunately, a matched cohort obtained using the same technology as IAR and PIT T cells is not available. We have acknowledged the potential bias introduced by using different TCR identification technologies in new text in the revised manuscript (lines 132-135 and 511-515).

7. The other avenue not explored by the authors is that the PIT-overlapping sequences might be a subset of IARs which is public and all their characteristics would be related to their public status, not to their IA reactivity.

*We have added a new figure showing significant overlap of public TCR junctions with PIT-matched (new **Figure S4**) and converged sequences (new **Figure S6**). We now state in the abstract that "PIT-matched TRA junctions were largely public and showed significant nucleotide sequence convergence..." (lines 29-31).*

8. By definition and experimental design, the TRAs shared between individuals - between IARs and PITs - are public sequences. Diverse metrics of PIT-overlapping TRAs described by authors might be universal characteristics of public sequences, not necessarily antigen-selection related. This is hinted at by PIT-matching sequences being shorter and closer to the germline sequences (line 341), suggesting that they might be easier to generate by VDJ recombination and hence be present in multiple individuals.

*We have added data showing Pgen scores (new **Figure S5**) in a section describing new data (lines 244-255). We conclude in the revised text that "Taken together, results from this section demonstrate that PIT-matched TCRs were enriched with public sequences, have high generational probability, and show TCR convergence" (lines 269-271). We believe this statement follows directly from the data and avoids the somewhat "Chicken or the Egg" problem making causal inferences about the different parameters.*

9. Besides of checking probability of generation of these TRAs, a good baseline would be overlap of non-antigen selected, TRA repertoires (and characteristics of overlapping versus nonoverlapping sequences) from individuals with matched HLA.

*We agree that this is a desirable aspirational goal. However, since such a repertoire is unavailable, we have addressed this problem using alternative approaches. We found that the overall HLA-dependence of PIT-matches did not significantly differ from the class II allele distribution in the total population (new **Figure S1**) (lines 121-124). We also found that variation in class II alleles did not greatly affect the extent of PIT matching in different TRA chain segments (moved from Supplemental Material to new **Figure 11**) (lines 197-202). Together, these findings suggest that peptide presentation by different PIT HLA molecules did not have a major effect on matching with IAR TRA junctions (lines 203-204). Finally, we have added a reference to HLA-independent TCR chain co-occurrence observed by others (lines 204-205).*

10. Also, the authors published previously on public sequences within IAR cells – comparison of the sequences identified in this work with the previously described would be of interest.

*As described in point 7 reviewer #1, we have added a new figure showing significant overlap of public TCR junctions with PIT-matched (new **Figure S4**) and converged sequences (new **Figure S6**). Our conclusion is that "Thus IAR TCRs, particularly public TRA junctions showed evidence of TCR convergence." (lines 268-269).*

11. I also could not find any supporting data within the main text nor figures for the abstract statement: "We detected extensive TCR sharing between IAR T cells from peripheral blood and pancreatic infiltrating T cells (PIT), with perfectly matched or single mismatched TRA junctions and J gene

regions, comprising ~34% of unique IAR TCRs."

Table 1 lists all perfectly TRA-matched sequences: there are 10 of them (in 47/1,606 cells, number of unique IAR TRAs not provided) - it is unclear where the number 34% comes from, especially as authors do not state what is the number of 0/1-mismatch sequences.

We apologize for the confusion. We have added more text explaining our calculations, and including the numbers of unique IAR TRAs in the revised manuscript (lines 159-161, 173-174 and 181-183).

12. Methods section is very scant and actually in many instances provides fewer details than the main text – e.g. the main text mentions magnetic separation of T cells, nowhere alluded to in the methods). Moving relevant details (like specific software used for string extraction/comparison) to the methods would make the paper more readable.

As requested, have moved numerous sentences throughout from the Results to the Supplemental Methods, and have also added additional text. We have highlighted changes in red font.

13. The paper would also benefit from shortening some parts, for example the reader does not need to know the path which lead the authors from comparison of TRA length to the comparison of V gene usage. A statement that TRA repertoires differ in their V gene usage and hence (because of V gene lengths) in TRA lengths would be easier to read.

We have extensively edited the revised text, removing excess verbiage. The specific change the reviewer requests was addressed in lines 331-333.

14. In summary, even though the dataset is unique and important for the field, I think this work requires major revision, more cautious data interpretation and a rewrite before publishing.

We appreciate the thoroughness of the reviewer's comments, particularly about T cell subtype and HLA class II allele usage of PIT-matches. We believe that by addressing these points, we have improved the revised manuscript.

Reviewer #2 (expert in type-1 diabetes immunopathogenesis):

Type 1 Diabetes is an organ-specific autoimmune disease, where many studies have shown that autoantigen-specific CD4 and CD8 T cells are involved. These cells recognize autoantigens through their T cell receptors (TCR) and, as such, understanding the nature of these TCRs is key to unravel the function of T cell in the pathogeny of the disease. Due to the complexities of sampling human pancreas, most studies have studied TCR from blood samples instead. Here, the authors go a step forward by analyzing TCR repertoires also in samples from human pancreas, finding that many TCRA chains shared by autoantigen-specific blood T cells and pancreas are of shorter length, more hydrophobic and potentially cross-reactive. While this approach is novel, the methodologies are appropriate, and the conclusions interesting, several issues remain:

Major concerns:

1. Page 5, lines 109-121: here the authors explain the sequencing of pancreatic infiltrating T cells (PIT), but there is no information about numbers of donors, demographic information, etc. The authors should include this information in the manuscript, as they did for other study participants.

*We have addressed the reviewer's points by adding to the revised manuscript a new **Table 1** and expanded **Table S1** that provide descriptions of the different cohorts and donors used in the study.*

2. Page 5, lines 124-125: here the authors explain how they did the matching of TCR sequences between Islet-antigen reactive CD4+ memory T cell (IAR) TCRs and PIT TCRs and indicate that sequence comparisons were made only at the amino acid level. To my opinion, this should be done also at the nucleotide level. Although this type of sharing is, in theory, less probable that perfect matches at amino acid level, the sharing of a TCR at the nucleotide level would be highly relevant, indicating identical thymic events between/among donors. Therefore, the analysis at the nucleotide level would add another level of significance to these results.

*We appreciate this comment and believe that in addressing it adds a new dimension to the manuscript. We have added an analysis of IAR TCR PIT-matching at the nucleotide level (new **Figure S6**). The data show significant TCR convergence (same junction amino acid sequence from multiple nucleotide sequences) (lines 256-271). We discuss evidence that TCR convergence has been associated with antigen-specificity (lines 464-467).*

3. Page 5, lines 128-129, and Page 7, line 172: although it becomes apparent that matching was higher for TRA, there was also matching for TRB. However, the authors only go on analysing [sic] TRA features- although understandable due to the higher effect observed, the fact of finding TRB sequences matching between IAR and PIT TCRs is an important finding and these matching TRB sequences should be further studied and described in the manuscript.

*We have added additional data on PIT TRB matches (new **Figures 1E and 1G, Figure S2E and 2G**) showing that they are less prevalent and less enriched in PIT matches than TRA junctions in both in Cohort 1 (**Figure 1** and **Figure S1**) and Cohort 2 (**Figure S2**). These figures are discussed in the revised text (lines 146-147, 161-163, 169-173). We have also added new data using TRB sequences for comparisons (new **Figures S4, S5 and S6**).*

4. Page 5, lines 129-131: if the referee understands this correctly, IAR cells are CD4+, while PIT cells were CD4 or CD8. If this is correct, how is it possible that many PIT CD8 TCRA sequences were found in IAR, CD4+, repertoires?

*We have added new data (new **Figures 1C and S1B**) as well as new text on sharing TCRs between CD4+ and CD8+ T cells (lines 121-124, lines 169-171). We have also added a reference that describes TCR junction overlap between CD4+ and CD8+ T cells (lines 115-117).*

5. Page 5, lines 132-133: when comparing TCRs from IAR and PIT, one would assume that if a given absolute number of IAR TCRs appear in PIT repertoires, the same absolute number of PIT TCRs would appear in IAR ones. However, these numbers are different in the text (47 vs 44). Could the authors clarify?

Thank you for pointing this out. We have corrected the numbers in the revised text (lines 111-113 and

elsewhere).

6. In general, the figures (main and supplementary) need to be improved for clarity, replacing “true” and “false” wording by the actual variable/group being shown (e.g. PIT-matching, non PIT-matching). This would ease the interpretation of the figures by the readers.

We have changed the PITmatch labels “TRUE/FALSE” to “PIT-matched and “non-PIT-matched” throughout the revised manuscript.

7. Pages 9-10, lines 248-252: PIT matched TRA junctions were shorter and more hydrophobic, while their paired TRB were not. What about the PIT matched TRB junctions?

*We have presented new data showing PIT-matched TRB junctions (new **Figures 1E, 1G, Figure S2E, G**). However, there were many more PIT-matched TRA than TRB junctions, making the former more highly powered for subsequent comparisons. For example, there were $n = 85$ perfect TRA junction matches for combined Cohorts 1 and 2, versus $n = 11$ perfect TRB matches. The situation was similar with single mismatched junctions (new **Figures 1A, and S2A**). Because of this disparity in numbers, and the increased power of comparisons with more junctions, we have elected to focus on more abundant TRA junctions (lines 179-181).*

8. Figure 3D: here the authors state that “Ratios of TRA CDR1 to CDR3 peptide contacts decreased significantly with increasing TRA junction length”. However, one could argue that, at longer lengths of CDR3, there are higher probabilities of having higher number of contacts, in which case both variables are related and it would be obvious that the ratio CDR1 contacts/CDR3 contacts would be lower if CDR3 length is higher. The authors should better explain what the novelty of this statement is.

*For clarification, we have re-written the revised text (lines 359-371). We have also modified the sentence in the legend to **Figure 3D** that is quoted by the reviewer (lines 859-860).*

9. Page 16, lines 445-447: “To help resolve this question, we show here that a significant fraction of IAR TCRs from peripheral blood share matching TRA chains with PIT TCRs, and vice versa”. The fraction of sharing is of 0.45% for PIT in IAR, and 2.9% for IAR in PIT (figures from Page 5, lines 132-133) which, although relevant and important, is considered not significant. The authors should tone down this claim.

We corrected this sentence in the revised manuscript (lines 454-456). This sentence now reads, “To help resolve this question, we show here that IAR CD4 T cell TCRs from peripheral blood share matching TRA chains, and lesser numbers of TRB chains, with PIT TCRs, and vice versa.”

10. Throughout the manuscript, on many occasions the authors mention results that are not shown. Given the importance of such statements for the overall message of the study, the following results should be shown/discussed, either as main or supplementary figures/tables, depending on space constraints and importance/relevance of the claims:

We have added several new Figures and have removed text references throughout that were not supported by data shown. More specific explanations are outlined in the comments below.

11. Page 5, lines 133-135: “The distribution of perfect matches between different subject groups (HC, T1D and newT1D) and cell types (CD4+, 135 CD8+) did not differ significantly from the distribution of total TCR populations from each group”.

*We have added new data (new **Figure S1**) showing distributions between HC, AAb+, and T1D groups in PIT-matches and total PIT cells (**Figure S1**). The distributions between matched and total sets did not differ significantly (lines 121-124).*

12. Page 8, lines 200-203: “We obtained results in Figures 1A-1E from Cohort 1 (Materials and Methods). To validate these findings, we repeated these analyses with all samples in an independent cohort (Cohort 2). We obtained essentially identical results, demonstrating that our observations were not restricted to a single data set and therefore had a potentially broader range of islet specificities”. This is particularly important- results for this cohort should be shown, instead of simply mentioning that they are “identical”.

*We have added new data (new **Figure S2**) repeating key comparisons in Cohort 2. We combined Cohorts 1 and 2 for subsequent comparisons (lines 176-178).*

13. Page 8, lines 210-212: “This preliminary analysis revealed that the fraction of PIT TRA junction matches with expanded IAR T cells was significantly elevated, and the fraction of PIT TRA junction non-matches reduced, in newT1D subjects, relative to HC and T1D subjects”.

We have removed this text from the revised manuscript.

14. Page 9, lines 239-242: “We found significant overlap between public IAR T cell and PIT-matched TRA junctions (p-value = 1.46×10^{-14} , hypergeometric distribution) but not private IAR T cell and PIT-matched TRA junctions (p-value > 0.05). Thus, there was strong overlap between cell populations with public and PIT-matching TCRs”.

*We have added new data (new **Figure S4**) showing the overlap between PIT-matched with public and private junctions. The numbers are slightly different because we used combined Cohorts 1 and 2 for this new comparison, but the conclusion is the same (lines 234-243).*

15. Page 11, lines 285-287: “In other analyses, we found that framework (FR) regions FR1, FR2, FR3 and FR4 did not differ in length between PIT-matched and non-matched TRA chains, showing selectivity of the differences in CDR1 and CDR3 lengths”.

*We have added new data showing the Framework region comparisons (new **Figures S8G-I**). The new data are discussed in the revised text (lines 319-321).*

Minor concerns and others:

16. Introduction: it should be mentioned that blood and pancreas samples are not from the same donors, for clarity.

We have made the requested changes in the revised text (lines 58-60).

17. Line 116: “represented 4,706 T cells”. Do the authors mean “4,706 CD4+ T cells”?

*We have removed this text from the revised manuscript and added the information to a new **Table 2**.*

18. Page 6, line 162: if here authors refer with “HC” to those donors from Su et al, it should be made clearer for the readers, as they could confuse these HC with those sequenced in the current study.

We have added text to the revised text indicating that the Su donors were uninfected, and that we refer to them as HC (line 131).

19. Page 8, line 209: has the size of down sampling been described in the methods section?

We have added a description of our down-sampling to the revised Supplemental Methods (lines 72-78).

20. Page 8, line 220: the sentence “in cells with expanded TCRs” is not clear, as a single cell can not have expanded TCRs. Do the authors mean expanded clones? If so, please rephrase for clarity.

We have corrected our wording in the revised text (line 219).

21. Figure 2, legend: it should explain what the brown histograms represent.

We have added this information in the revised text (line 841-842).

22. Pages 11-12, lines 300-311: this paragraph could be moved after line 282, as it would ease the understanding of these section of the results.

*We would agree with the reviewer if we were preparing a verbal presentation. However, this move would place discussion of **Figure S5** in the middle of the discussion of **Figure S4**. This would be contrary to standard scientific writing and would, we believe, create even more confusion. We would, therefore, prefer to keep the order of the paragraphs the same unless directed otherwise by the editor.*

23. Pages 13-14, lines 365-366: do the authors mean “non-matching” or “single mismatching”?

We have clarified the revised text (lines 378).

24. Page 14, line 379: the sequence of these multi-specific TCR clones is not shown; given the importance of these results, they should be shown, for example in Table 3.

*We have added the sequences of the multi-specific TCR clones to **Table 4**. These tables are discussed in the revised text (lines 210-211).*

25. Page 14, lines 389-390: “TCR transduced primary CD4+ T cells”. The methods for the generation of these cells are not shown.

We have added a description to the revised Supplemental Methods (lines 80-90)

26. Page 15, line 414: it should read "greater".

We have corrected this error in the revised text.

27. Page 18, line 500: it should read "utilize".

We have corrected this error and moved this section to Supplemental Methods (line 111).

28. Supplementary file, Page 3, lines 38-40: "Cohort 2 samples were stimulated with overlapping peptide libraries from the same islet proteins optimized for HLA class II presentation (20 aa, 12 aa overlap, Table S3)". The table does not seem to be included.

*We apologize for this this omission. We have added this table (now **Table S4**) to the Supplemental Tables.*

REVIEWERS' COMMENTS

Reviewer #1 (Remarks to the Author):

Authors compared TCR receptors of islet antigen reactive cells (IARs) in individuals with active T1D autoimmunity/healthy with those found in pancreatic infiltrations of T1D-affected and unaffected individuals.

Authors found extensive sharing of TRA chain (in contrast to TRB chain) between these two types of repertoires. Shared sequences had characteristics previously linked to crossreactivity (partially also tested/modelled by authors) and suggest that such crossreactive T cells are present in the pancreas, enabling/helping autoimmune process.

This paper is a valuable contribution in the field, especially because of the data from the T1D-target tissue and offers a link between pre-existing TCR repertoire and autoimmunity ; I recommend it for publication.

Nevertheless, it would benefit from some changes:

Discussion, Line 460 - describes increase of PIT-matching TRA chains in blood in time leading to diagnosis (corresponding text in the Results section 214-228 and the figure 1K). The figure suggests (and authors state it in the result section) that the comparison of expanded clonotypes was inconclusive, as underpowered; yet in the discussion they write 'We also show that frequencies of PIT-matching TRA chains in blood increase prior to the time of diagnosis, suggesting a temporal linkage of levels of PIT-matched TRA chains in blood with disease progression.' - requires toning down.

A point to add to the discussion (as it immediately raises interest when reading) TRAV41*01 genes - found previously in innate cells (line 330) - could authors elaborate/speculate on cells with these TRAs in the discussion?

Another point for the discussion: due to the leakiness of allelic exclusion, a substantial fraction of T cells has two recombined TRA chains, lower (but still relevant) fraction expresses both alleles. How would it matter for the mechanisms suggested by authors?

Line 401 "In parallel and independent experiments, we unexpectedly found that the TRA chain of a TCR (P196-1) from Influenza A/MP54- reactive" - it is unclear to me whether these experiments are a part of this paper, if yes, where are they described (I assume it is the a CFSE proliferation assay fig 4, but can't find description?

Minor issues:

line 105 "string matches " - unnecessary/technical detail, methods or remove altogether

line 178 'analyses' would be more appropriate than experiments?

fig. S7A and text line 294: could authors use IMGT numbering, broadly used in the field? Also in other places throughout the text?

Line 303 "analogous to immunoglobulin" - unnecessary

Fig 2. The figure would be more immediately legible if TRA and TRB panels were marked on the figure

Fig5 legend title NoEpitopes a bit misleading, - consider also a common legend to save space

Reviewer #1 (Remarks on code availability):

The code repository is extensive, contains code to reconstruct paper figures and comprises the data used for analyses (I have not verified whether this is complete data). Authors should be praised for their transparent approach. For an interested reader, it would be very helpful if they would structure the repository more and would add appropriate readme files and files extensions (.R).

Reviewer #2 (Remarks to the Author):

The authors have now successfully addressed the concerns raised during the first review. A few new minor issues appear after reviewing the new version of the manuscript:

- Lines 46-48 of Supplemental Methods and 83-85 of main text: in this new version the authors describe more deeply the methods employed. In particular, they state that "IAR CD4+ T cell TCRs with in frame protein sequences were selected and filtered by removing iNKT cells with the CVVSDRGSTLGRLYF junction; and MAIT cells with the TRAV1-2 V gene and TRAJ33, TRAJ20 or TRAJ12 J genes." Was this also done for PIT TCRs? How many sequences were removed from each cell type and TCR subset (IAR,PIT)?
- Figure 1B: isn't this figure redundant with regards to Figure 1A? If 55 TCRA PIT sequences are perfectly shared with IAR, it is obvious than the opposite is also true.
- Line 164 of main text: do authors refer to Tables 2 and 3, or 3 and 4?
- The titles of Tables 3 and 4 are not sufficiently clear to the reader: does Table 3 refer to known or unknown specificities? Why in this table the number of TRA is not 55? What about unknown specificities with single mismatches?
- Figure S2B not being significant, while Fig1F is. Do the authors have an explanation for this cohort-specific effect?
- Figure 1 caption needs to be clearer to indicate which plots refer to Cohort 1, and which ones refer to combined Cohort 1+2 results.
- Line 241: the authors state that "Neither public nor private IAR TRB junctions showed overlap with PIT-matched TRB junctions (Figure S4C-D)." However, data in S4C is significant- therefore, public TRB junctions do show a significant overlap with PIT-matched TRB junctions.
- Line 256: potential typo "Public TCR AA sequences TCRs may arise from".
- Lines 322-323: the results above showed that non-template nucleotide regions are also different. The conclusion as it stands indicates that only regions in the V gene are different.
- Line 340: which was the basis to select those 30 TCRs for further study? Known antigen specificity?
- Line 528: study ID for iReceptor database is missing.
- Line 101 of supplementary methods: potential typo. Do the authors mean "Junctions having >1 nt sequence per AA sequence were designated as "Converged"?"
- Some figure captions do not indicate the statistical test used.
- Figure S13: how is it possible that 97% of live lymphocytes are CD4+? One could guess that this might be due to CD154 enrichment prior to flow cytometry- this should be explained in the figure caption to make it clear to the readers.

RESPONSE TO REVIEWERS' COMMENTS

We thank the reviewers for their thorough and careful reviews throughout the review process. We believe the revisions we have made in response to the reviewer's comments have greatly improved our manuscript. Here, we have revised the manuscript again addressing each of the new comments.

Reviewer #1 (Remarks to the Author):

1. Discussion, Line 460 - describes increase of PIT-matching TRA chains in blood in time leading to diagnosis (corresponding text in the Results section 214-228 and the figure 1K). The figure suggests (and authors state it in the result section) that the comparison of expanded clonotypes was inconclusive, as underpowered; yet in the discussion they write 'We also show that frequencies of PIT-matching TRA chains in blood increase prior to the time of diagnosis, suggesting a temporal linkage of levels of PIT-matched TRA chains in blood with disease progression.' - requires toning down.

As requested, we have toned down the discussion of PIT-matching TRA chains in blood prior to diagnosis (lines 477-480).

2. A point to add to the discussion (as it immediately raises interest when reading) *TRAV41*01* genes - found previously in innate cells (line 330) - could authors elaborate/speculate on cells with these TRAs in the discussion?

We have added a sentence to the discussion (lines 499-501) speculating on TRAV41 T cells

3. Another point for the discussion: due to the leakiness of allelic exclusion, a substantial fraction of T cells has two recombined TRA chains, lower (but still relevant) fraction expresses both alleles. How would it matter for the mechanisms suggested by authors?

We have added a sentence to the discussion (lines 501-503) suggesting future studies on the relationship between PIT-matched TCRs and dual TRA TCRs.

4. Line 401 "In parallel and independent experiments, we unexpectedly found that the TRA chain of a TCR (P196-1) from Influenza A/MP54- reactive" - it is unclear to me whether these experiments are a part of this paper, if yes, where are they described (I assume it is the a CFSE proliferation assay fig 4, but can't find description?

We have added text describing the discovery and cloning of this TCR (lines 608-617) and a reference to this description Main text, (line 418).

Minor issues:

5. line 105 "string matches " -unnecessary/technical detail, methods or remove altogether

We have removed this phrase (line 113).

6. line 178 'analyses' would be more appropriate than experiments?

We have made this modification (line 191).

7. fig. S7A and text line 294: could authors use IMGT numbering, broadly used in the field? Also in other places throughout the text?

We appreciate the reviewer's desire for nomenclature consistency with IMGT. However, it is difficult to compare absolute positions of junctions between different TCRs. Different V genes vary in length, causing the beginning of the junction region to occur over a range of distances from the N terminus that vary by as much as ~14 amino acids, a distance as great as the total length of many junctions. We therefore find it easier to compare relative distances from the beginning of the junction. To avoid confusion, we have changed the X axis label on Figure S7A from "Junction (AA)" to "Position (AA relative to Junction start)".

8. Line 303 "analogous to immunoglobulin" - unnecessary

We have removed this phrase (line 317).

9. Fig 2. The figure would be more immediately legible if TRA and TRB panels were marked on the figure

We have added TRA and TRB labels to Figure 2, as requested. To further clarify, we have edited the figure legend (line 1011).

10. Fig5 legend title NoEpitopes a bit misleading, - consider also a common legend to save space

To avoid confusion, we have changed the legend title in Figure 5 from "NoEpitopes" to "No. Epitopes". We have also added a common legend, as requested.

11. The code repository is extensive, contains code to reconstruct paper figures and comprises the data used for analyses (I have not verified whether this is complete data). Authors should be praised for their transparent approach. For an interested reader, it would be very helpful if they

would structure the repository more and would add appropriate readme files and files extensions (.R).

We have added a README file describing the structure of the repository (<https://github.com/BenaroyaResearch/Germline-like-TCR-alpha-chains-shared-between-autoreactive-T-cells-in-blood-and-pancreas>). We have also changed the code file extensions to “.R”, as requested.

Reviewer #2 (Remarks to the Author):

12. - Lines 46-48 of Supplemental Methods and 83-85 of main text: in this new version the authors describe more deeply the methods employed. In particular, they state that “IAR CD4+ T cell TCRs with in frame protein sequences were selected and filtered by removing iNKT cells with the CVVSDRGSTLGRLYF junction; and MAIT cells with the TRAV1-2 V gene and TRAJ33, TRAJ20 or TRAJ12 J genes.” Was this also done for PIT TCRs? How many sequences were removed from each cell type and TCR subset (IAR,PIT)?

For IAR T cells, we removed iNKT cells with the CVVSDRGSTLGRLYF junction (n = 21 cells, 47 junctions); and MAIT cells with the TRAV1-2 V gene and TRAJ33, TRAJ20 or TRAJ12 J genes (n = 18 cells, 45 junctions). We have added this information to the Methods (lines 771-774). Although PIT TCRs also contained iNKT and MAIT sequences (~1.6% total junctions), we did not also remove iNKT and MAIT cell TCRs from PIT TCRs because we felt it was redundant after removal of these sequences from IAR TCRs. As expected, removal of the iNKT and MAIT cell TCRs from PIT TCRs did not affect numbers of IAR matches.

13. - Figure 1B: isn't this figure redundant with regards to Figure 1A? If 55 TCRA PIT sequences are perfectly shared with IAR, it is obvious than the opposite is also true.

We understand the reviewer's point but feel that including a description of the reciprocal sharing provides a sanity check for our procedures. To avoid confusion, we have added the text “As expected” to the section describing these results (line 121). If requested by the editor, we would be happy to remove the figure,

14.- Line 164 of main text: do authors refer to Tables 2 and 3, or 3 and 4?

*Thank you for catching this error. We have corrected the text to refer to **Tables 3 and 4** (lines 173-178; see also response #15 below).*

15. - The titles of Tables 3 and 4 are not sufficiently clear to the reader: does Table 3 refer to known or unknown specificities? Why in this table the number of TRA is not 55? What about unknown specificities with single mismatches?

*We are grateful to the reviewer for pointing this out. To clarify, we have made changes to the title (line 949) and legend (lines 950-952) to **Table 3** that the table shows selected examples (n = 10 out of 55 total) of TCRs with unknown specificity. We have also re-written the description of **Tables 3 and 4** in the main text (lines 173-178). We do not feel that adding information on unknown specificities with single mismatches will contribute to the main conclusion we derived from these tables, namely that paired TRB junctions were markedly more divergent than matched TRA junctions (lines 176-178).*

16- Figure S2B not being significant, while Fig1F is. Do the authors have an explanation for this cohort-specific effect?

Thanks to the reviewer for pointing out these apparent cohort-specific effects. While we do not fully understand the basis of the differences, we suspect they are related to the smaller sample size in Cohort 2 (Figure 2SF and G) relative to Cohort 1 (Figure 1F and G). Rather than adding more text attempting to explain something we don't understand, we elected to take a simpler approach of removing the two panels, which we feel are unnecessary, from the Supplemental Figures.

17. - Figure 1 caption needs to be clearer to indicate which plots refer to Cohort 1, and which ones refer to combined Cohort 1+2 results.

We have added a clarifying sentence to the legend of Figure 1 (lines 978-979).

18. - Line 241: the authors state that “Neither public nor private IAR TRB junctions showed overlap with PIT-matched TRB junctions (Figure S4C-D).” However, data in S4C is significant- therefore, public TRB junctions do show a significant overlap with PIT-matched TRB junctions.

We have corrected this statement in the revised text (lines 256-257).

19. - Line 256: potential typo “Public TCR AA sequences TCRs may arise from”.

We have changed the word “arise” to “result”, which we believe to be a more appropriate word choice (line 259).

20. - Lines 322-323: the results above showed that non-template nucleotide regions are also different. The conclusion as it stands indicates that only regions in the V gene are different.

We have expanded the scope of this statement in the revised text (lines 337-339).

21.- Line 340: which was the basis to select those 30 TCRs for further study? Known antigen specificity?

We have clarified the revised text (lines 357-359).

22.- Line 528: study ID for iReceptor database is missing.

We have added the iReceptor Study ID (line 878).

23. - Line 101 of supplementary methods: potential typo. Do the authors mean “Junctions having >1 nt sequence per AA sequence were designated as “Converged”?”

We have modified this sentence for clarity (lines 679-681), Methods).

24. - Some figure captions do not indicate the statistical test used.

We have added information on statistical tests used to the figure legends throughout the revised Main text and Supplemental Figures.

25.- Figure S13: how is it possible that 97% of live lymphocytes are CD4+? One could guess that this might be due to CD154 enrichment prior to flow cytometry- this should be explained in the figure caption to make it clear to the readers.

*The reason why the CD3+ cells are 97% CD4+ in Figure S13 is that an anti-CD8 antibody was included in our dump channel (Table S4). We have modified the legend for **Figure S13** to include this information as the reviewer requested.*